# Origin and evolution of transporter substrate specificity within the NPF family

Morten Egevang Jørgensen[1,2], Deyang Xu[1,2], Christoph Crocoll[1,2], Heidi Asschenfeldt Ernst[3], David Ramírez[4,5], Mohammed Saddik Motawia[6], Carl Erik Olsen[7], Osman Mirza[3], Hussam Hassan Nour-Eldin[1,2]*, Barbara Ann Halkier[1,2]*

[1]DynaMo Center, Department of Plant and Environmental Sciences, Faculty of Science, University of Copenhagen, Frederiksberg, Denmark; [2]Copenhagen Plant Science Center, Department of Plant and Environmental Sciences, Faculty of Science, University of Copenhagen, Frederiksberg, Denmark; [3]Department of Drug Design and Pharmacology, Faculty of Health and Medical Sciences, University of Copenhagen, Copenhagen, Denmark; [4]Centro de Bioinformatica y Simulacion Molecular (CBSM), Universidad de Talca, Talca, Chile; [5]Instituto de Innovación Basada en Ciencia, Universidad de Talca, Talca, Chile; [6]Center for Plant Plasticity, Department of Plant and Environmental Sciences, Faculty of Science, University of Copenhagen, Frederiksberg, Denmark; [7]Department of Plant and Environmental Sciences, Faculty of Science, University of Copenhagen, Frederiksberg, Denmark

*For correspondence:
huha@plen.ku.dk (HHN-E);
bah@plen.ku.dk (BAH)

**Abstract** Despite vast diversity in metabolites and the matching substrate specificity of their transporters, little is known about how evolution of transporter substrate specificities is linked to emergence of substrates via evolution of biosynthetic pathways. Transporter specificity towards the recently evolved glucosinolates characteristic of *Brassicales* is shown to evolve prior to emergence of glucosinolate biosynthesis. Furthermore, we show that glucosinolate transporters belonging to the ubiquitous NRT1/PTR FAMILY (NPF) likely evolved from transporters of the ancestral cyanogenic glucosides found across more than 2500 species outside of the *Brassicales*. Biochemical characterization of orthologs along the phylogenetic lineage from cassava to *A. thaliana*, suggests that alterations in the electrogenicity of the transporters accompanied changes in substrate specificity. Linking the evolutionary path of transporter substrate specificities to that of the biosynthetic pathways, exemplify how transporter substrate specificities originate and evolve as new biosynthesis pathways emerge.
DOI: https://doi.org/10.7554/eLife.19466.001

## Introduction

Phospholipid-based cell membranes are the foundation for extant cellular life and with them arose the need for carrier proteins to shuttle metabolites across the semi-permeable membranes (*Mansy et al., 2008*; *Deamer and Dworkin, 2005*). New biosynthesis pathways continuously emerged throughout evolution, resulting in a vast diversity in metabolite chemical structures (>200.000 structures tentatively identified in the plant kingdom alone), where some are restricted to certain taxa and others are found broadly (*Weng et al., 2012*). Matching the vast structural diversity of metabolites, approximately 10% of coding sequences of contemporary genomes encode transport proteins with diverse substrate specificities (*Saier and Ren, 2006*) that enable transport of metabolites and ions into and out of cells. However, the evolutionary path that leads to the rise of new transporter substrate specificity upon emergence of new metabolites is unknown. Classical

**eLife digest** All living cells are surrounded by membranes that protect them from the external environment. The membrane contains proteins called transporters, which move nutrients and other molecules (known as substrates) across the membrane. A variety of transporters have evolved to move the hundreds of thousands of different substrates found in nature.

Plant cells make many different compounds to protect themselves from pests and diseases. A group of transporters known as the NPF family move some of these compounds across the cells outer membrane. The types of substrates they transport vary in different plants. In cassava, for example, NPF transporters move compounds called cyanogenic glucosides, which are poisonous to humans and other animals. On the other hand, NPF transporters in another plant called *Arabidopsis thaliana* can move bitter-tasting compounds called glucosinolates. The process that makes glucosinolates in plants evolved from the process that makes cyanogenic glucosides.

Can transporters evolve the ability to move a new substrate before or after that substrate first appears? To answer this question, Jørgensen et al. studied the NPF family in *A. thaliana*, cassava and another plant called papaya that makes both cyanogenic glucosides and glucosinolates. The experiments suggest that NPF transporters able to move both cyanogenic glucosides and glucosinolates evolved before plants evolved the ability to make glucosinolates. Later in evolution, these multi-specific transporters specialized to only move glucosinolates. Jørgensen et al. also show that early glucosinolate transporters could move a broad variety of glucosinolates but later evolved to only transport particular types.

These findings show how transporters and the processes that make compounds in cells may evolve together. A future challenge will be to understand the molecular changes in a transporter that make it specific for a certain substrate. This may help researchers to develop new ways of controlling the amount of toxic compounds in crops we eat by manipulating how the compounds are transported.

DOI: https://doi.org/10.7554/eLife.19466.002

evolution theory (*Ohno, 2013*) and several studies (e.g. *Fani and Fondi, 2009*; *Prasad et al., 2012*) support the hypothesis that new enzyme functions arise in duplicated genes if they are subject to unique selection pressure, - alternatively they rapidly become pseudogenes. A key example is found in the evolution of mineral corticoid and glucocorticoid receptors found in vertebrates (*Bridgham et al., 2006*; *Carroll et al., 2008*). These two receptors evolved post duplication of a dual specificity aldosterone and cortisol receptor basal to the jawed vertebrate lineage (*Bridgham et al., 2006*). However, aldosterone biosynthesis did not arise before the advent of tetrapods suggesting that the ancestral receptor evolved affinity towards aldosterone before the hormone was present, possibly as a by-product of the receptors' affinity towards chemically similar ligands (*Bridgham et al., 2006*; *Carroll et al., 2008*). Thus, it appears that selection pressure enforced upon related but distinct ligands can drive the emergence of receptors' affinity towards a new substrate. In comparison, it is not clear if new transporters evolve de novo with emergence of new substrates, or whether gene duplications allow ancestral multifunctional proteins to take on greater specificity (*Khersonsky and Tawfik, 2010*).

To answer this question, it is necessary to use a model system where the evolution of the biosynthetic pathway is known and where transporters have been identified. As a model system, we used the *Brassicales*-specific glucosinolate defense compounds with a biosynthetic pathway that diversified from the ancestral cyanogenic glucoside pathway found in more than 2500 plant species (*Sønderby et al., 2010a*; *Halkier and Gershenzon, 2006*; *Bak et al., 1998*; *Clausen et al., 2015*; *Mithen et al., 2010*). The two pathways share the initial enzymatic amino acid to oxime conversion, but produce structurally different end products (*Clausen et al., 2015*). Through an arms race between plants and interacting organisms (*Bidart-Bouzat and Kliebenstein, 2008*; *Züst et al., 2012*; *Kliebenstein et al., 2005*; *de Vos et al., 2008*; *Sanchez-Vallet et al., 2010*; *Prasad et al., 2012*; *Newton et al., 2009*; *Fahey et al., 2001*; *Agerbirk and Olsen, 2012*), the glucosinolate pathway evolved to produce >130 glucosinolate structures with diverse amino acid-derived side chains (*Fahey et al., 2001*; *Agerbirk and Olsen, 2012*). Also, two $H^+$/glucosinolate symporters, GTR1 and

GTR2, belonging to the NPF family (*Léran et al., 2014*) and with broad glucosinolate specificity (i.e. no discrimination against amino acid side chain) were identified in *Arabidopsis thaliana* (*Nour-Eldin et al., 2012*) that predominantly produces aliphatic and indole glucosinolates (*Mithen et al., 2010*; *Brown et al., 2003*). Although transporters for cyanogenic glucosides are yet to be identified (*Jørgensen et al., 2005*), we set out to investigate whether the evolution of a new biosynthetic pathway (here glucosinolates from cyanogenic glucosides) promoted the co-evolution of transporter specificity, i.e. did glucosinolate transporters originate from cyanogenic glucoside transporters in the NPF family? Furthermore, *in planta* studies suggest the existence of an additional glucosinolate transporter with narrow specificity for the recently evolved indole glucosinolates (*Andersen et al., 2013*) that are essential for innate immune responses (*Sanchez-Vallet et al., 2010*; *Clay et al., 2009*; *Bednarek et al., 2009*). We therefore investigated if evolution within a biosynthetic pathway (here emergence of indole glucosinolates) is accompanied by evolution in transporter substrate specificity.

Here we identify the first cyanogenic glucoside transporter in cassava and the first indole-specific glucosinolate transporter in *A. thaliana*. By characterizing substrate specificity and electrogenicity in orthologs along the phylogenetic lineage from cassava to *A. thaliana*, we provide a model for the evolutionary path of the substrate specificity of a plant specialized metabolite transporter. Surprisingly, we show that glucosinolate transport capacity likely occurred prior to the emergence of glucosinolate biosynthesis in dual-specificity transporters of cyanogenic glucosides and glucosinolates. With the emergence of glucosinolate biosynthesis, the transporters lost the capacity to transport cyanogenic glucosides. Moreover, we show that the first glucosinolate transporters had broad specificity and later subfunctionalized towards specific classes of glucosinolates. Our data suggests that changes in electrogenicity have accompanied the evolutionary changes in substrate specificity. Our results exemplify how new transporter substrate specificities evolve when new metabolites arise.

## Results and discussion

### Identification of an indole-specific glucosinolate transporter

To assess the evolutionary path of GTR transporters, we first set out to identify the putative indole-specific glucosinolate transporter. In a previous study (*Nour-Eldin et al., 2012*), we found that the glucosinolate transport capability of the NPF family is confined to the NPF2.8–2.14 transporters that cluster closely with AtGTR1 (NPF2.10) and AtGTR2 (NPF2.11). We thus hypothesized that the indole-specific glucosinolate transporter could be found in this NPF subclade in *A. thaliana* (*Figure 1A*). Via heterologous expression in *Xenopus laevis* oocytes, we screened six of the seven members within this subclade for transport of indol-3-yl-methyl glucosinolate (I3M, the simplest indole glucosinolate) and 4-methylthiobutyl glucosinolate (4MTB) – representing a highly abundant aliphatic glucosinolate in *A. thaliana* (*Figure 1A–B*). NPF2.9 (At1g18880, hereafter GTR3) - the closest homolog of GTR1 and GTR2 - transported I3M effectively (*Figure 1C*). Two Electrode Voltage Clamp (TEVC) electrophysiology and time-course uptake assays showed that I3M, but not 4MTB, induces negative currents in GTR3-expressing oocytes (*Figure 2A–B*) and that GTR3 can over-accumulate I3M, but not 4MTB, against a concentration gradient (*Figure 2C–D* and *Figure 2—figure supplement 1A–B*). Alternatively, an un-coupled conductance may accompany 4MTB transport in GTR3 resulting in non-electrogenic transport or transport rates may be below the electrophysiological detection level. In comparison, GTR1 over-accumulated both 4MTB and I3M (*Figure 2C–D* and *Figure 2—figure supplement 1A–B*) and elicited negative currents of similar amplitude for both glucosinolates (*Figure 2A–B*).

Plotting currents at $-60$ mV as a function of increasing I3M concentrations yielded a saturation curve best fitted by a Michaelis-Menten equation with $K_m$ towards I3M <25 uM for GTR1, GTR2 and GTR3 (*Figure 2E–F* and *Figure 2—figure supplement 2*). Through competition assays we show that GTR3-mediated 4MTB uptake is strongly inhibited by 10% I3M, whereas 10-fold excess 4MTB does not affect I3M uptake (*Figure 3A–D*). In contrast, GTR1 transports 4MTB and I3M to the same ratio as applied in the assay media (*Figure 3A–D*). In accordance with previous characterization (*Wang and Tsay, 2011*), GTR3 imports nitrate into oocytes (*Figure 3G*). Nitrate at concentrations 100-fold in excess of I3M or 4MTB did not outcompete uptake of neither glucosinolate, indicating that the two substrates are not mutually exclusive (*Figure 3E–G*). In conclusion, our biochemical

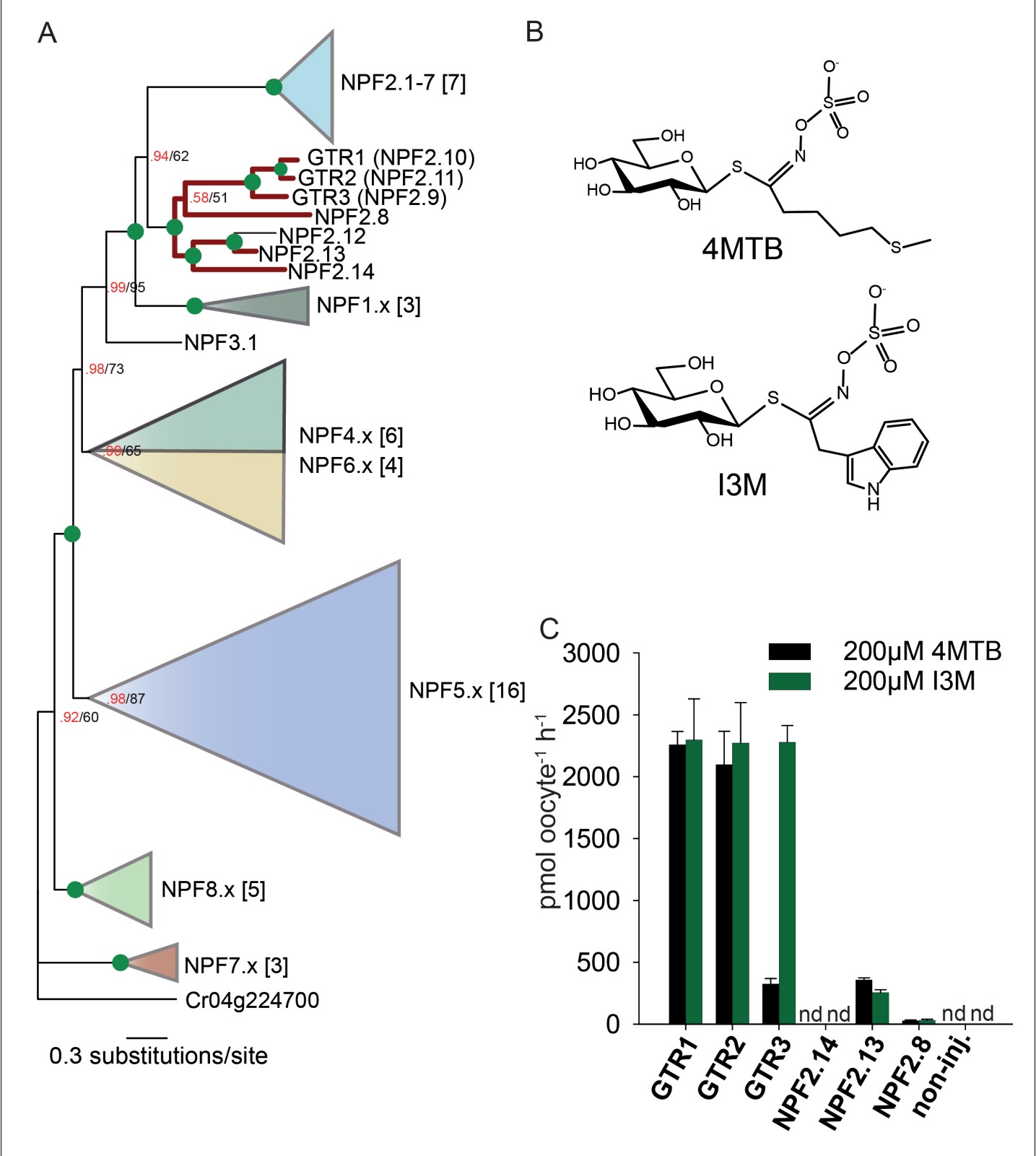

**Figure 1.** Identification of the indole-specific glucosinolate transporter GTR3 in the NPF family. (**A**) Bayesian inference (MrBayes) tree (s.d.< 0.01) of the *A. thaliana* NPF family with reduced phylogenies of NPF1.x, NPF3.x, NPF4.x, NPF5.x, NPF6.x, NPF7.x, NPF8.x and NPF2.1–7 clades (x denotes the subfamily number) as previously annotated (*Léran et al., 2014*). Numbers in brackets indicate the number of genes in reduced phylogeny. Green circles at nodes represent a posterior probability of 1 (maximum is 1). Values at nodes separated by a backslash represent MrBayes values below 1 in

*Figure 1 continued on next page*

*Figure 1 continued*

red, followed by RAxML generated bootstrap values in black (only reported when Mrbayes value is below 1). GTR1, GTR2, GTR3 and 3 other homologs tested in **B**) are highlighted with red branches. For non-reduced phylogeny, see *Figure 1—figure supplement 1*. (**B**) The chemical structure of 4-methylthiobutyl glucosinolate (4MTB) and indol-3-ylmethyl glucosinolate (I3M). (**C**) GTR1, GTR2, GTR3, NPF2.14, NPF2.13 and NPF2.8 were expressed individually in 15 *X. laevis* oocytes and transport activity was measured in the presence of 0.2 mM 4MTB (black bars) or 0.2 mM I3M (green bars). 4MTB or I3M accumulated within oocytes were quantified by LC-MS analyses in 3 × 5 oocytes for each gene. Error bars represent ± s.d. n = 3, experiment repeated two times; nd=none detected.
DOI: https://doi.org/10.7554/eLife.19466.003
The following figure supplement is available for figure 1:

**Figure supplement 1.** Non-reduced AtNPF tree.
DOI: https://doi.org/10.7554/eLife.19466.004

characterization shows that GTR3 is an electrogenic transporter with a high apparent affinity and strong preference for indole glucosinolates.

We investigated the physiological relevance of GTR3′s role as an indole glucosinolate transporter *in planta*. GTR3 is strongly expressed in the plasma membrane of root phloem companion cells (*Wang and Tsay, 2011*) and is co-expressed with GTR1 and GTR2 at the tissue level in other tissues according to publicly available translatome data (*Mustroph et al., 2009*) (*Figure 4—figure supplement 1*). *A. thaliana gtr3* mutants accumulate significantly lower concentrations of indole glucosinolates in roots compared to wild type (*Figure 4A*). This shift was increased in *gtr1 gtr2 gtr3* triple knock-out (tko), but was not seen in *gtr1 gtr2 double knock-out* (dko) (*Figure 4A–B*, [*Andersen et al., 2013*]). In the rosette, there is a trend, but no statistically significant increase in indole glucosinolates of the *gtr3* mutant when compared to wild type (*Figure 4B*). The *gtr1 gtr2* dko shows a statistically significant increase in the rosette levels of indole glucosinolate that is further increased to 4 fold when also knocking out GTR3 (*gtr1 gtr2 gtr3* tko) (*Figure 4B*). This suggests that GTR1, GTR2 and GTR3 all contribute to distributing indole glucosinolates between root and shoot. We used micro-grafting to further investigate the role of GTR1, GTR2 and GTR3 in the source-sink relationship for indole glucosinolates between root and rosette. As MYB28 and MYB29 – key regulators of aliphatic glucosinolate biosynthesis (*Sønderby et al., 2010b*) – are not necessary for expression of GTR1, GTR2 and GTR3 (*Müller et al., 2010*), we could use the glucosinolate biosynthetic null mutant - *myb28/myb29 cyp79b2/cyp79b3* quadruple knockout (qko) - in micro-grafting experiments. By micro-grafting four-day-old *A. thaliana* seedlings of qko, *gtr1 gtr2 gtr3* tko and wild type plants we created reciprocal grafts of roots and rosettes from all genotypes and analyzed glucosinolate content in root and rosette of three-week-old grafted plants.

Based on substrate-specificity and overlapping expression of GTR1, GTR2 and GTR3 we would expect that the distribution of aliphatic glucosinolates in a *gtr1 gtr2 gtr3* tko would resemble the pattern in *gtr1 gtr2* dko plants. In agreement, distribution of aliphatic glucosinolates in heterografts of *gtr1 gtr2 gtr3* tko with wild type and qko plants, respectively, showed similar changes in distribution pattern for aliphatic glucosinolates as previously reported for heterografts of *gtr1 gtr2* dko with wild type and qko plants, respectively (*Figure 4—figure supplement 2* and [*Andersen et al., 2013*]). Furthermore, the grafting procedure does not influence the glucosinolate distribution as evidenced by homografts of wild type and *gtr1 gtr2 gtr3* tko plants showing a similar distribution of indole glucosinolates as seen for non-grafted plants (*Figure 4C*), and by homografts of qko plants being devoid of indole glucosinolates (*Figure 4C* and [*Andersen et al., 2013*]). Analysis of heterografted plants with no glucosinolate biosynthesis in the root showed that only small amounts of indole glucosinolates are transported from rosette to root. Similarly, it was evident from the (qko/wt) heterografts that when all three GTRs are expressed in the root, the root to rosette transport is below detection levels (*Figure 4C*). However, when all three GTRs are knocked out in roots, we see a dramatic increase in the rosette indole glucosinolate content (*Figure 4*). In combination, this suggests that GTR3 (along with GTR1 and GTR2) has a role in retaining indole glucosinolates in the root, presumably by importing indole glucosinolates into storage cells.

## Rise and evolution of glucosinolate transport specificity

From an evolutionary perspective, our findings propose two models for how substrate specificity evolved for the glucosinolate transporters. Either glucosinolate transport first arose with narrow

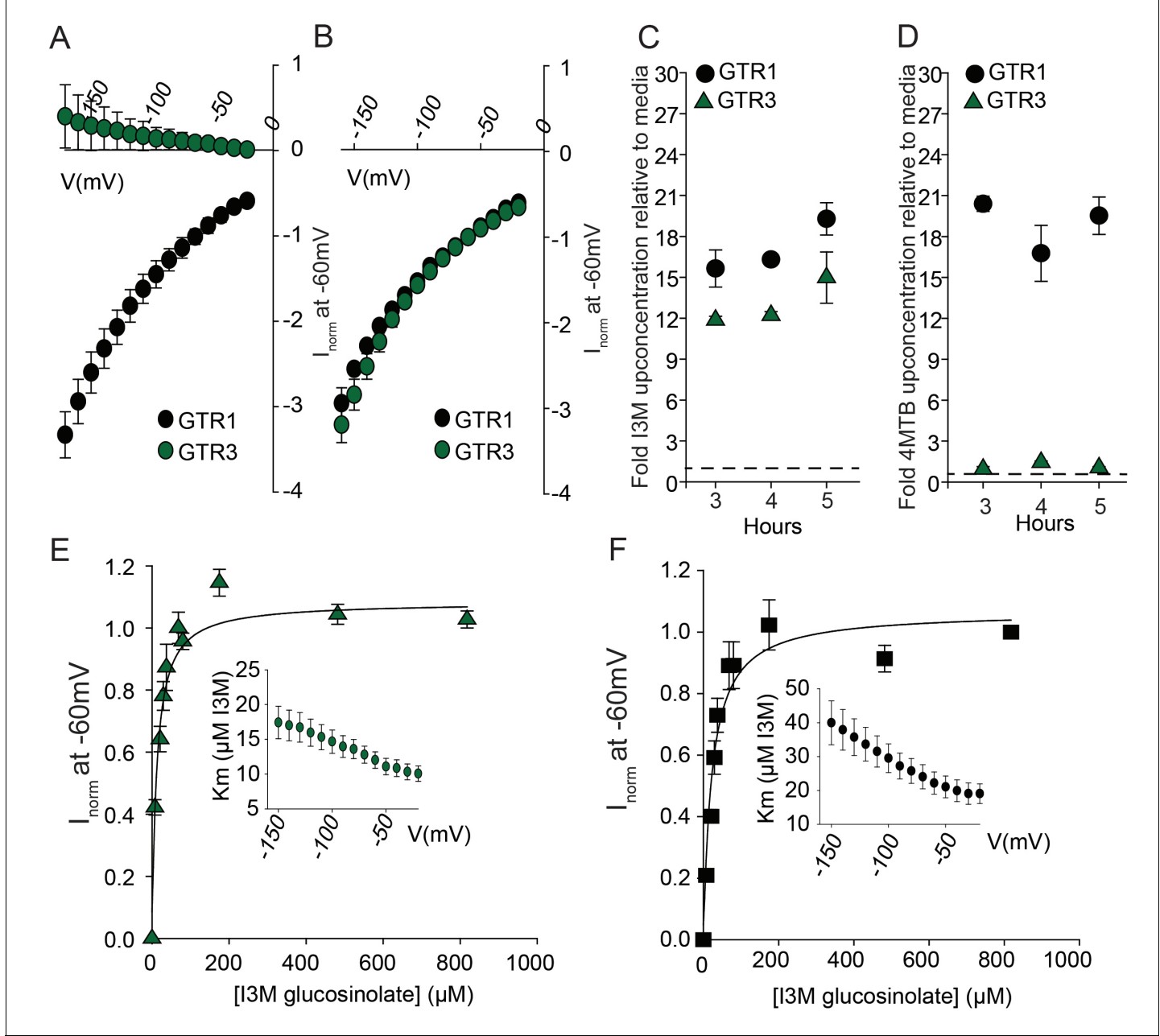

**Figure 2.** Biochemical characterization of the indole-specific glucosinolate transporter GTR3. (A–B) Normalized IV (Current-Voltage) curve of 4MTB (**A**)- and I3M (**B**)-induced currents for GTR1 (black circles)- and GTR3 (green circles)-expressing oocytes exposed to 100 µM substrate at pH5. Both GTR1 and GTR3 currents were normalized to GTR1 currents elicited at saturating 4MTB concentrations and at a membrane potential of −60 mV (Error bars represent ± s.e., n = 6, experiment repeated two times). (**C–D**) Time-dependent accumulation of I3M (**C**) and 4MTB (**D**), respectively, relative to assay media concentration in GTR1- and GTR3-expressing oocytes. Accumulated 4MTB or I3M were quantified by LC-MS in 3 × 5 oocytes for each gene after 3, 4 and 5 hr of incubation in a standard pH5 Kulori buffer containing 0.2 mM I3M or 0.2 mM 4MTB (error bars represent ± s.d. n = 3). Dotted line represents media concentration. (**E–F**) Normalized I3M-induced currents for GTR3 (**E**) or GTR1 (**F**) measured at a membrane potential of −60 mV and pH 5 plotted against increasing I3M concentrations. The saturation curve was fitted with a Michaelis-Menten equation represented by a solid line. Each oocyte dataset was normalized to currents elicited at 0.8 mM I3M concentration at −60 mV. The insert shows the apparent $K_m$ as a function of membrane potential. Error bars represent ± s.e. of mean, n = 6, experiment repeated two times.

DOI: https://doi.org/10.7554/eLife.19466.005

The following figure supplements are available for figure 2:

**Figure supplement 1.** Uptake of 4MTB and I3M by GTR1 and GTR3 expressed in *X. laevis* oocytes.

DOI: https://doi.org/10.7554/eLife.19466.006

*Figure 2 continued on next page*

*Figure 2 continued*

**Figure supplement 2.** GTR2 indole glucosinolate $K_m$ measurement.
DOI: https://doi.org/10.7554/eLife.19466.007

specificity for indole glucosinolates followed by a broadening of the substrate specificity, or the reverse. To address this question and potentially determine when glucosinolate transport capability arose, we performed a phylogenetic analysis of NPF transporters from glucosinolate-producing species (*A. thaliana*, *Brassica rapa* and *Carica papaya*) and non-producing species (*Theobroma cacao* (cacao), *Manihot esculenta* (cassava), *Glycine max*, *Gossypium raimondi* and *Solanum lycopersicum*). The phylogenetic analyses revealed three well-defined *AtGTR1-*, *AtGTR2-* and *AtGTR3*-containing subclades with NPF sequences exclusively from glucosinolate-producing species of the *Brassicaceae* (*A. thaliana* and *B. rapa*) (*Figure 5A* and *Figure 5—figure supplement 1*). Additionally, the analyses revealed a subclade, which grouped basal to the three GTR1–3 subclades. This subclade, which we name the GTR-like subclade contained GTR homologs from *C. papaya*, the most basal glucosinolate-producing species in *Brassicales* with a sequenced genome (*Mithen et al., 2010*; *Goodstein et al., 2012*), and from non-producing species (*M. truncatula*, *G. raimondii*, *S. lycopersicum*, *M. esculenta* and *T. cacao*). The four subclades grouped in a larger clade, which we name the GTR-clade.

To track the rise and evolution of glucosinolate substrate specificity we tested a range of transporters for glucosinolate transport activity via expression in *X. laevis* oocytes. All transporters described below were codon optimized for expression in *X. laevis* oocytes and tested for transport activity in their native form (i.e. without tag). Additionally, we fused each gene to YFP in the C-terminus and confirmed its expression and localization to the plasmamembrane via confocal microscopy (*Figure 5—figure supplement 2*). In the following, lack of transport can therefore likely be attributed to lack of activity rather than lack of expression. Within the GTR1 and GTR3 subclades, we tested one of the respective orthologs from *B. rapa* and showed a strong preference for I3M by the tested BrGTR3 ortholog (BrH02396), whereas the tested BrGTR1 ortholog (BrF01711) transported 4MTB and I3M with similar efficiency (*Figure 5B*). Thus, the high preference for indole glucosinolates appears typical for GTR3 orthologs within the *Brassicales*-specific GTR3 subclade. Oocytes expressing GTR-like transporters from *C. papaya* over-accumulate both 4MTB and I3M relative to the assay media concentration and we named them *CARICA PAPAYA* Glucosinolate Transporter LIKE-1 (CpGTRL1) and −2 (CpGTRL2), respectively (*Figure 5B*). The ability of CpGTRL1 and −2 to transport both 4MTB and I3M is surprising as indole glucosinolates are found in *A. thaliana* and *B. rapa*, but not in *C. papaya* (*Mithen et al., 2010*). Interestingly, CpGTRL2 from C. papaya transported 4MTB with a $k_m$ of 85 ± 12 µM at −60 mV (*Figure 5D*). This indicates that the high affinity of GTRs towards glucosinolates evolved before the diversification of the *Brassicaceae* and *Caricaceae*. Moreover, the data imply that the common ancestor of the GTR transporters was originally broad-specific and that GTR3 lost the ability to over-accumulate aliphatic glucosinolates after the divergence of *C. papaya* and the ancestor of *Arabidopsis* and *Brassica* (~72.1 MYA, median of 8 studies [*Hedges et al., 2006*]). This suggests that preference for indole glucosinolates evolved as a subfunctionalization of ancestral, broad-specific glucosinolate transporters.

To track the rise of transport capacity towards glucosinolates, we tested for glucosinolate uptake in the closest GTR homologs from cassava, which do not produce glucosinolates but produces the evolutionary related and ancestral cyanogenic glucosides (*McMohan et al., 1995*). As a control, we included GTR homologs from cacao, which produces neither compound class (*Bjerg et al., 1987*; *Seigler, 2005*). Oocytes expressing Me14G074000 from cassava over-accumulated both 4MTB and I3M relative to external media, while the expressed GTR homolog from cacao did not transport any glucosinolates (*Figure 5B*, *Figure 5—figure supplement 3* and *Figure 5—figure supplement 2*). Hence, as cassava does not synthesize glucosinolates, the ability to transport glucosinolate appears to have arisen in the NPF family prior to the evolution of the glucosinolate biosynthetic pathway. Furthermore, the characterization of this potentially ancestral form of the glucosinolate transporters support that they first evolved with broad specificity towards aliphatic and indole glucosinolates.

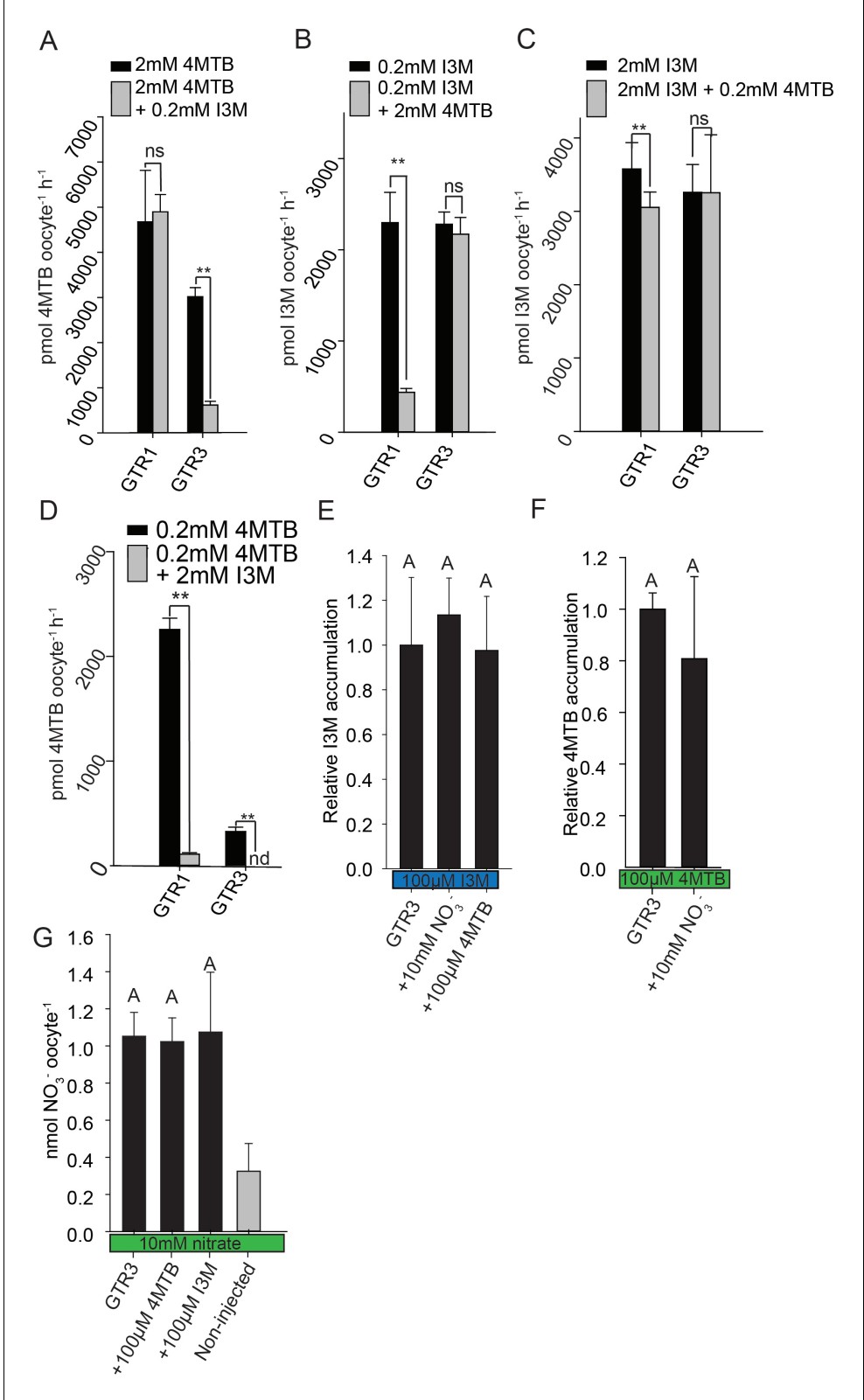

**Figure 3.** Substrate competition assays of GTRs in *X. laevis* oocytes. (**A–D**) Competition for uptake of I3M and 4MTB into oocytes expressing GTR1 or GTR3. (**A**) Quantification of 4MTB uptake when oocytes were exposed to high 4MTB concentration (2 mM) alone or in combination with low concentration of I3M (0.2 mM). (**B**) Quantification of I3M uptake when low I3M concentration (0.2 mM) was competed with high concentration of 4MTB (2 mM). (**C**) Quantification of I3M when oocytes were exposed to high I3M concentration (2 mM) alone or in combination with low concentration of
*Figure 3 continued on next page*

*Figure 3 continued*

4MTB (0.2 mM). (**D**) Quantification of 4MTB uptake when oocytes were exposed to low I3M concentration (0.2 mM) alone or in combination with high concentration of 4MTB (2 mM). Accumulated 4MTB (**A** and **D**) or I3M (**B** and **C**) was quantified by LC-MS in $3 \times 5$ oocytes for each gene. Two tailed T-test, \*\*p<0.001 vs non-competed, \*p<0.05 vs non-competed. NS= not significantly different (Error bars represent ± s.d. of mean for data obtained from three times five different oocytes per experiment). (**E**–**G**) Quantification of nitrate and glucosinolate competition assays. (**E**) Quantification of I3M uptake in GTR3-expressing oocytes when saturating I3M concentration (0.1 mM) is competed with high concentration of $NO_3^-$ (10 mM) or saturating concentration of 4MTB (0.1 mM). (**F**) Quantification of 4MTB uptake in GTR3-expressing oocytes when saturating 4MTB concentration (0.1 mM) is competed with high concentration of $NO_3^-$ (10 mM). (**G**) Quantification of $NO_3^-$ uptake in GTR3-expressing oocytes when high concentration of $NO_3^-$ (10 mM) is competed by 0.1 mM 4MTB or saturating concentration of I3M (0.1 mM). Accumulated I3M (**E**) or 4MTB (**F**) was quantified by LC-MS in $3 \times 5$ oocytes for each gene. Accumulated $NO_3^-$ (**G**) was quantified by ICP-MS in three oocytes for each gene. Error bars represent ± s.d. of mean, n = 3. Groups in subfigures are determined by one-way ANOVA followed by Holm-Sidak post-hoc analysis (p<0.05).

DOI: https://doi.org/10.7554/eLife.19466.008

## Co-evolution of transporter substrate specificity and substrate biosynthesis

Glucosinolates and cyanogenic glucosides are structurally distinct (*Figure 1B* and *Figure 6A*), yet they share functional moieties (glucose moiety and amino acid-derived side chains) (*Bak et al., 1998*; *Clausen et al., 2015*). Based on this and the existence of a transporter in cassava with glucosinolate transport capacity, we explored whether the glucosinolate NPF transporters may have evolved from cyanogenic glucoside transporters within the NPF family, much like glucosinolates evolved from cyanogenic glucoside biosynthesis (*Bak et al., 1998*; *Clausen et al., 2015*; *Mithen et al., 2010*). To test this hypothesis we investigated the ability of selected GTRs and GTR-like homologs to transport representative cyanogenic glucosides, namely phenylalanine-derived prunasin and valine-derived linamarin (*Figure 6A*). Notably, cassava produces only cyanogenic glucosides whereas *C. papaya* produces both cyanogenic glucosides and glucosinolates (derived from phenylalanine). Oocytes expressing the glucosinolate transporters from *C. papaya*, *B. rapa* and *A. thaliana* did not accumulate the tested cyanogenic glucosides above trace amounts (*Figure 6A*). In comparison, oocytes expressing Me14G074000 from cassava accumulated prunasin, but not linamarin, to levels equivalent to media (*Figure 6A* and *Figure 6—figure supplement 1*). This indicates that the substrate-binding cavity of this NPF transporter can accommodate both cyanogenic glucosides and glucosinolates. As Me14G074000 transports both compound classes, we propose that this transporter may represent a transition phase where specificity for cyanogenic glucosides is partially lost in favor of glucosinolate transport. This suggests that glucosinolate transporters evolved from those of cyanogenic glucosides.

If Me14G074000 represents such a 'transition' transporter between cyanogenic glucoside-specific to glucosinolate-specific transporter, we hypothesized that the genome of the cyanogenic glucoside-producing cassava also encodes a GTR-like NPF transporter that is specific for cyanogenic glucosides. We tested this hypothesis by measuring transport activity of the six closest homologs of Me14G074000 from cassava (*Figure 6B–C* and *Figure 6—figure supplement 2*). The closest homolog, Me14G074100, appears truncated (data not shown) with only five transmembrane-spanning domains but was nevertheless included in our analysis. All six transporters were tested for transport activity in their native form. Additionally, we fused YFP to the C-terminus of each gene to validate expression. Native Me14G074100, Me01G191900, Me09G097200, and Me17G124600 did not result in uptake of 4MTB, I3M, prunasin or linamarin. Among the YFP-tagged transporters, only Me14G074100 and Me09G097200 did not express in the oocytes (*Figure 6—figure supplement 1*), and hence we cannot conclude whether these two transporters are inactive. Uptake of prunasin was detected in oocytes expressing Me15G176100 at levels similar to or slightly lower than the suggested 'transition' transporter (Me14G074000) (*Figure 6C*). In contrast, Me15G180400 strongly over-accumulated prunasin and linamarin to more than 12 and 8 times the media level (*Figure 6C*) while uptake of both aliphatic and indole glucosinolates by this transporter was negligible (*Figure 6—figure supplement 3*). Thus, Me15G180400 is specific towards cyanogenic glucosides. We named Me15G180400 MANIHOT ESCULENTA CYANOGENIC GLUCOSIDE TRANSPORTER-1 (MeCGTR1) and to the best of our knowledge, it represents the first identification of an importer of cyanogenic glucosides. TEVC electrophysiology assays showed that prunasin and linamarin induce negative currents in MeCGTR1-expressing oocytes (*Figure 6D*). Kinetic analysis of MeCGTR1

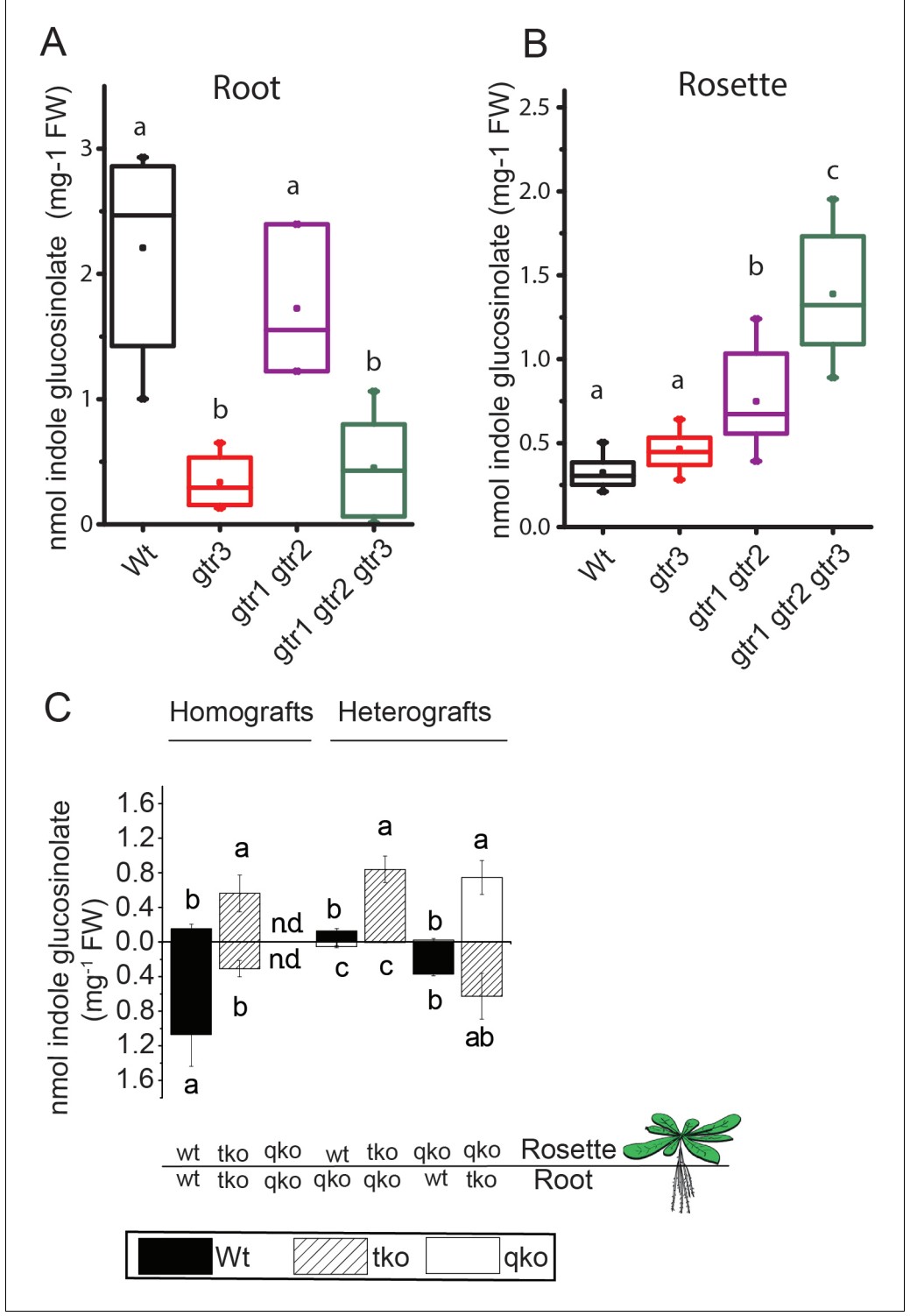

**Figure 4.** *In planta* characterization of the indole-specific glucosinolate transporter GTR3. (A–B) Indole glucosinolate content in (A) root and (B) rosette of non-grafted hydroponically grown wildtype, *gtr3, gtr1 gtr2* dko and *gtr1 gtr2 gtr3* tko plants. The box is determined by the 25th and 75th percentiles. The whiskers are determined by the 5th and 95th percentiles. Median and mean are shown as line and square. Groups in subfigures are determined by one-way ANOVA followed by Tukey HSD Calculator multiple comparison post-hoc analysis (p<0.05). Data presented is one of two individual experiments, each containing 8–12 repeats (n) (see *Figure 4— source data 1–2* for individual glucosinolate data points and individual n; error bars and parentheses are s.d. (C) Indole glucosinolate concentrations of micro-grafted 3-week-old plate-grown *Arabidopsis* wildtype (Col-0) and

*Figure 4 continued on next page*

*Figure 4 continued*

mutants. Rosettes and roots from wild type (wt), the glucosinolate biosynthesis *null* mutant *myb28 myb29 cyp79b2 cyp79b3* (qko) and the *gtr1 gtr2 gtr3* mutant (tko) were reciprocally grafted using 4-day-old seedlings. Glucosinolate content in the rosette and roots was quantified by LC-MS in 3-week-old plants. Data presented is one of two individual experiments, each containing 8–16 repeats (n) (see *Figure 4—source data 3–4* for individual glucosinolate data points and individual n; error bars and parentheses are s.d. Groups in subfigures are determined by one-way ANOVA (p<0.05). n.d. none detected.

DOI: https://doi.org/10.7554/eLife.19466.009

The following source data and figure supplements are available for figure 4:

**Source data 1.** Glucosinolate content in rosettes of hydroponically grown wildtype, *gtr3*, *gtr1 gtr2 dko* and *gtr1 gtr2 gtr3 tko* plants.
DOI: https://doi.org/10.7554/eLife.19466.013

**Source data 2.** Glucosinolate content in roots of hydroponically grown wildtype, *gtr3*, *gtr1 gtr2 dko* and *gtr1 gtr2 gtr3 tko* plants.
DOI: https://doi.org/10.7554/eLife.19466.014

**Source data 3.** Glucosinolate content in rosettes of micro-grafted plants.
DOI: https://doi.org/10.7554/eLife.19466.015

**Source data 4.** Glucosinolate content in roots of micro-grafted plants.
DOI: https://doi.org/10.7554/eLife.19466.016

**Figure supplement 1.** *In silico* expression analysis of the indole-specific glucosinolate transporter GTR3 and the broad-specificity GTR1 and GTR2 transporters.
DOI: https://doi.org/10.7554/eLife.19466.010

**Figure supplement 2.** *In planta* characterization of the indole-specific glucosinolate transporter GTR3 – aliphatic glucosinolate data.
DOI: https://doi.org/10.7554/eLife.19466.011

**Figure supplement 3.** Validation of *gtr3* T-DNA insertion mutants.
DOI: https://doi.org/10.7554/eLife.19466.012

---

showed that this protein transports prunasin and linamarin with a $K_m$ of $80 \pm 7$ μM and $262 \pm 15$ μM, respectively, measured at a membrane potential clamped to $-60$ mV (*Figure 6E–F*). This indicates that MeCGTR1 is a high-affinity, cyanogenic glucoside-specific transporter and shows that it is capable of over-accumulating against a concentration gradient. The existence of MeCGTR1 supports our hypothesis that the dual-specific Me14G074000 represents a 'transition' transporter evolutionarily positioned between cyanogenic glucoside-specific (MeCGTR1) and glucosinolate-specific transporters (GTR1–3 and GTRL1–2). The identification and close phylogenetic relationship of glucosinolate-specific, dual-specific and cyanogenic glucoside-specific transporters within the NPF supports that glucosinolate transporters evolved from cyanogenic glucoside NPF transporters.

## Is evolution of new substrate specificity in the NPF accompanied by changes in transporter electrogenicity?

Most characterized members of the SLC15/PepT/POT/NPF family are symporters that function by an electrogenic proton-coupled transport mechanism (*Nour-Eldin et al., 2012*; *Parker and Newstead, 2014*; *Fei et al., 1999*; *Chen et al., 1999*; *Steel et al., 1997*; *Mackenzie et al., 1996*; *Fei et al., 1994*; *Doki et al., 2013*; *Solcan et al., 2012*; *Chiang et al., 2004*), that is, symport of protons generates a net influx of positive charge that can be measured as a negative current by TEVC. Characterization of the transporters identified in this study by both LCMS- and TEVC-based transport assays enabled us to investigate the evolution of electrogenicity of glucosinolate and cyanogenic glucoside transporters. Previously, we showed that AtGTR1 and −2 mediated transport of 4MTB induces negative currents as a result of net inward movement of protons during transport (*Nour-Eldin et al., 2012*). In this study, we show that negative currents are also induced by both AtGTR1 and −2 when exposed to I3M (*Figure 2B* and *Figure 2—figure supplement 2*). This indicates that AtGTR1 and −2 transport 4MTB and I3M – two negatively charged glucosinolates with different amino acid side chains- via a similar electrogenic transport mechanism. Similarly, the tested GTR1 ortholog from *B. rapa* (BrGTR1) also induced negative currents when exposed to 4MTB or I3M (*Figure 5C*). In comparison, AtGTR3 and the GTR3 ortholog from *B. rapa* (BrGTR3) only induced currents when exposed to I3M (*Figure 5C*). No detectable currents were induced by 4MTB in neither

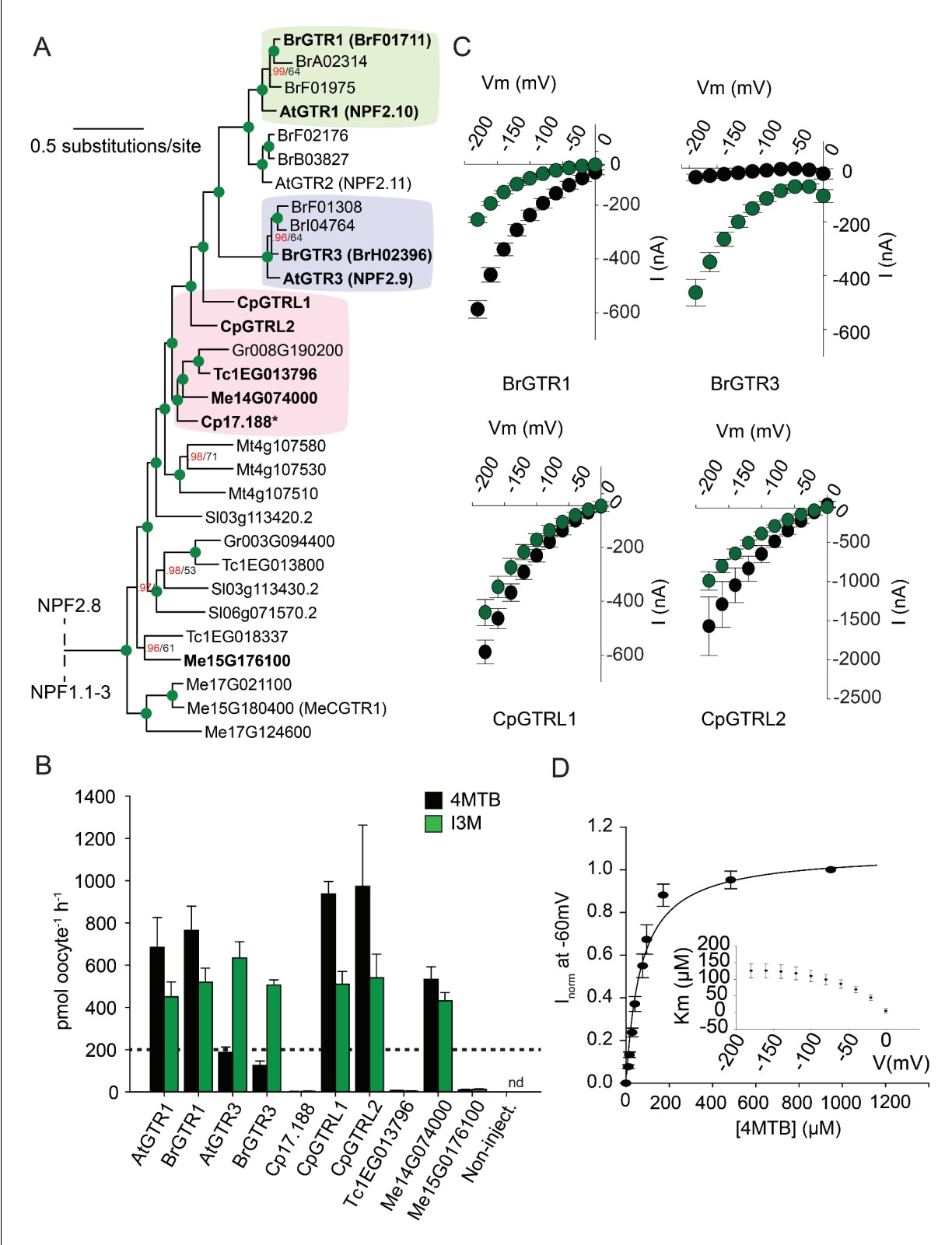

**Figure 5.** Phylogenetic relationship and transport specificity of GTR homologs from selected species. (A) Selected part of Bayesian inference (MrBayes) tree (s.d. < 0.01) of GTR homologs from selected species (Full phylogenetic tree of NPFs from selected species is found as *Figure 5—figure supplement 1*). Green circles at nodes represent a posterior probability of 1 (maximum is 1). Values at nodes separated by a backslash represent MrBayes values below 1 in red, followed by RAxML generated bootstrap values in black (only reported when Mrbayes value is below 1). Asterix

*Figure 5 continued on next page*

Figure 5 continued

indicates that *Cp17.188* lacks the highly conserved EXXE[R/K] motif involved in proton-coupling (*Jørgensen et al., 2015*). Subclades with green, purple and pink background denote the GTR1 subclade, GTR3 subclade and GTR-like subclade (genes that cluster with GTR homologs from *C.papaya*), respectively. Genes in bold were tested for glucosinolate transport in B). (B) Uptake of 4MTB and I3M by *X. laevis* oocytes expressing selected GTR homologs (bold) from *A. thaliana*, *B. rapa*, *C. papaya*, *T. cacao* and *M. esculenta* from the colored subclades and Me15G176100, which clusters outside the GTR-like subclade). Genes were expressed individually in *X. laevis* oocytes and transport activity was measured in the presence of 0.2 mM 4MTB (black bars) or 0.2 mM I3M (green bars) at external pH 5. Dotted line represents substrate concentration in external media. Accumulated 4MTB or I3M was quantified by LC-MS in 5 × 1 oocytes for each gene (Error bars represent ± s.d. of mean, n = 5, experiment repeated two times). (C) 4MTB (black circles)- and I3M (green circles)-induced currents in oocytes expressing GTR homologs that showed glucosinolate uptake in B). Expressing oocytes were exposed to 0.2 mM 4MTB or I3M and induced currents were measured at membrane potentials clamped between 0 mV and −180 mV in 20 mV increments at pH 5 (Error bars represent ± s.d. of mean, n = 4, experiment repeated two times). (D) Normalized 4MTB-induced currents of CpGTRL2 (Cp17.190) measured at a membrane potential of −60 mV at pH 5 plotted against increasing 4MTB concentrations. The saturation curve was fitted with a Michaelis-Menten equation represented by a solid line (Error bars represent ± s.d. of mean for data obtained from four different oocytes per experiment). Each oocyte dataset was normalized to currents elicited at 1 mM 4MTB concentration at −60 mV. The insert shows the apparent $K_m$ as a function of membrane potential.

DOI: https://doi.org/10.7554/eLife.19466.017

The following figure supplements are available for figure 5:

**Figure supplement 1.** Phylogenetic relationships of NPF transporters from selected species.

DOI: https://doi.org/10.7554/eLife.19466.018

**Figure supplement 2.** TEVC electrophysiology measurements of AtGTR1, AtGTR3 and Me15G74000.

DOI: https://doi.org/10.7554/eLife.19466.019

**Figure supplement 3.** Expression analysis of YFP-tagged (C-terminal) GTRs and GTR homologs from *B. rapa*, *C. papaya*, *T. cacao* and cassava in *X. laevis* oocytes.

DOI: https://doi.org/10.7554/eLife.19466.020

AtGTR3 nor BrGTR3 (*Figure 2A–B* and *Figure 5C*). Thus, it appears that electrogenic transport of - and the ability to upconcentrate - 4MTB is a property that distinguishes the GTR1 clade from the GTR3 clade. Similarly, exposure to cyanogenic glucosides did not induce currents in the putative transition transporter, Me14G074000; (*Figure 6D*). In comparison, MeCGTR1 induced negative currents when exposed to the non-charged prunasin and linamarin (*Figure 6D*). This suggests that transport of these two cyanogenic glucosides by MeCGTR1 is coupled to a net influx of cationic species and that transport of cyanogenic glucosides by MeCGTR1 and Me14G074000 appears to differ with respect to electrogenicity.

Identification of the phylogenetically more basal glucosinolate transporters in *C. papaya* and cassava allowed us to investigate when electrogenic glucosinolate transport may have evolved. From the GTR-like clade, the glucosinolate transporting CpGTRL1 and CpGTRL2 induced currents when exposed to 4MTB or I3M (*Figure 5C*) whereas Me14G074000 from cassava did not induce detectable currents (*Figure 5—figure supplement 3C*). All three transporters were able to upconcentrate both glucosinolates against their respective concentration gradient (*Figure 5B*). Thus, our data suggest that the glucosinolate transport mechanism first arose as a non-electrogenic mechanism that later evolved to become electrogenic. Moreover, electrogenic transport appears not to be a prerequisite for the ability to over-accumulate glucosinolates. Previously, we showed that negative currents induced by AtGTR1 and AtGTR2 when exposed to the negatively charged glucosinolates, reflect a glucosinolate to proton stoichiometry of 1 ≤ 2 (*Nour-Eldin et al., 2012*). Substrate-dependent variation in transport coupling stoichiometry between substrate and protons has been shown to depend on the length of the oligo-peptide substrate for PepT$_{So}$ (*Parker and Newstead, 2014*). The non-electrogenic transport by Me14G074000 could indicate a different proton to glucosinolate stoichiometry compared to that of the *A. thaliana* orthologs. For example, the lack of detectable currents may be caused by an equal amount of negative and positive charges moving across the membrane during the transport cycle. This would suggest that changes in transporter substrate specificity for a given substrate are accompanied by changes in transporter electrogenicity. However, we cannot exclude that the lack of currents for Me14G074000 is caused by currents below detection limits or that co-transport of other ions may be ´masking´ the coupled transport by Me14G074000. Nevertheless, the absence of induced currents by Me14G074000 indicates that transport of glucosinolates became electrogenic after the divergence of cassava and *C. papaya*.

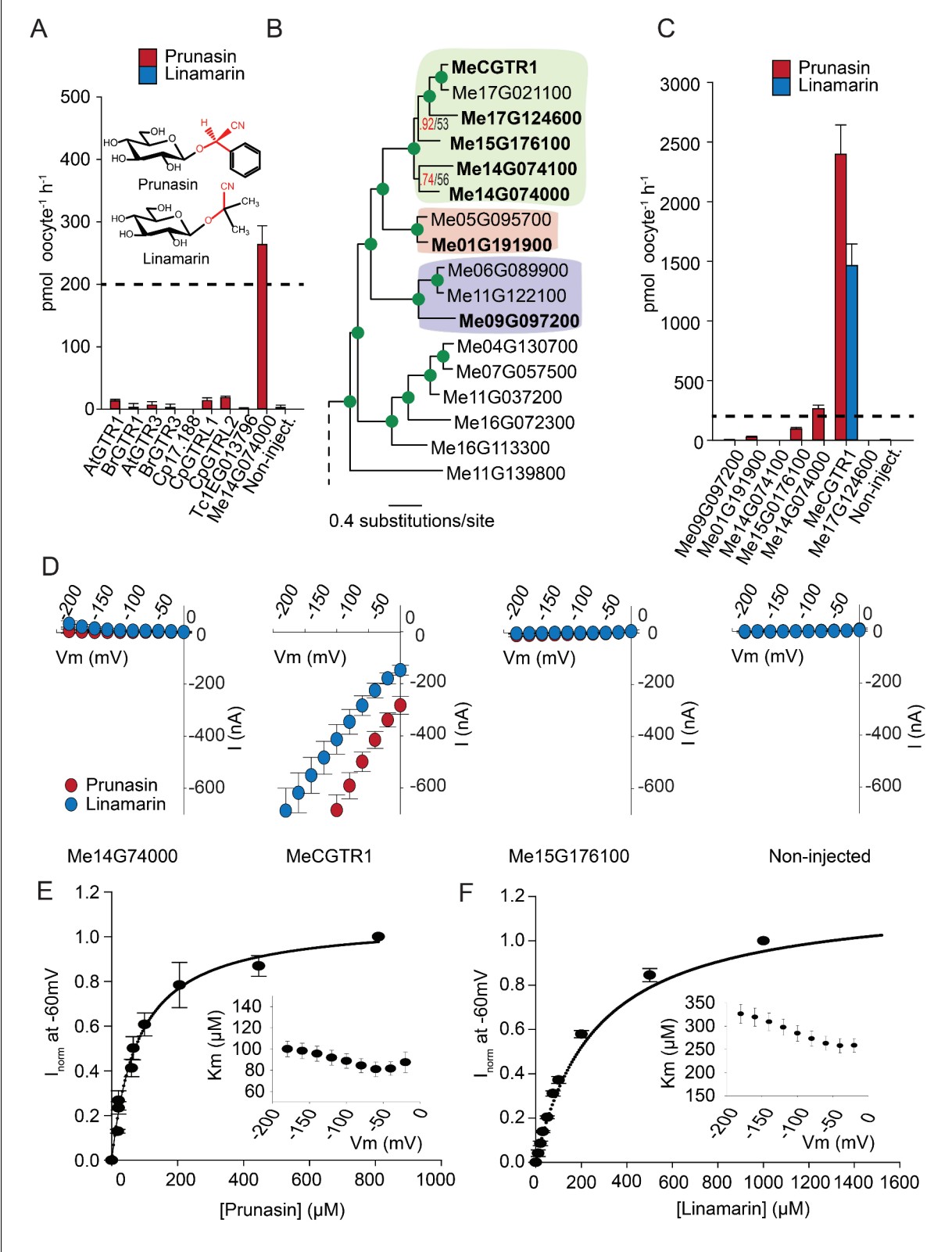

**Figure 6.** Biochemical characterization of cyanogenic glucoside NPF transporters. (**A**) Uptake of prunasin and linamarin in *X. laevis* oocytes expressing GTR homologs from *A. thaliana*, *B. rapa*, *C. papaya*, *T. cacao* and *M.esculenta*. Genes were expressed individually in *X. laevis* oocytes and transport activity was measured in the presence of 0.2 mM prunasin (red bars) or linamarin (blue bars). Accumulated prunasin or linamarin were quantified by LC-MS in 5 × 1 oocytes for each gene (Error bars represent ±s.d. of mean for data obtained from five different oocytes per experiment, experiment

*Figure 6 continued on next page*

*Figure 6 continued*

repeated two times). Dotted line represents media substrate concentration. None of the genes accumulated linamarin to detectable levels. (B) Bayesian inference tree (MrBayes tree) showing selected part of *M. esculenta* NPF phylogenetic tree (closest homologs of *Me14G074000)*. Green circles at nodes represent a posterior probability of 1 (maximum is 1). Values at nodes separated by a backslash represent MrBayes values below 1 in red, followed by RAxML generated bootstrap values in black (only reported when Mrbayes value is below 1). Scale bar indicates number of substitutions per site. Subclades coloured green, pink and purple mark genes that cluster with GTRs, NPF2.12/13 and NPF2.8, respectively, in *Figure 5—figure supplement 1*. Full phylogenetic tree of *M.esculenta* NPFs is found as *Figure 6—figure supplement 2*. Genes in bold were assayed for prunasin, linamarin, 4MTB and I3M uptake. (C) Accumulation of prunasin and linamarin in *X.laevis* oocytes expressing closest homologs of *Me14G074000* from *M. esculenta*. Genes were expressed individually in *X. laevis* oocytes and transport activity was measured in the presence of 0.2 mM prunasin (red bars) or 0.2 mM linamarin (blue bars). Accumulated prunasin or linamarin was quantified by LC-MS in 5 × 1 oocytes for each gene (Error bars represent ± s.d. of mean for data obtained from five different oocytes per experiment, experiment repeated two times). Only MeCGTR1 accumulated linamarin to detectable levels. Dotted line represents substrate concentration in external media. (D) Prunasin (red circles)- and linamarin (blue circles)-induced currents in oocytes expressing Me14G74000, MeCGTR1, Me15G176100 and non-expressing oocytes, respectively. Expressing and non-expressing oocytes were exposed to 0.2 mM prunasin or 0.2 mM linamarin and induced currents were measured at membrane potentials between 0 mV and −180 mV in 20 mV increments at pH5. (E–F) Normalized prunasin (E) or linamarin (F) induced currents elicited in MeCGTR1-expressing oocytes measured at a membrane potential of −60 mV and pH 5 plotted against increasing prunasin (E) or linamarin (F) concentrations. The saturation curve was fitted with a Michaelis-Menten equation represented by a solid line (error bars are s.d.; n = 4). Each oocyte dataset was normalized to currents elicited at 0.8 mM prunasin (E) or 1 mM linamarin (F) concentration at −60 mV. The insert shows the apparent $K_m$ as a function of membrane potential (error bars are s.d.; n = 3–4 oocytes).

DOI: https://doi.org/10.7554/eLife.19466.021

The following figure supplements are available for figure 6:

**Figure supplement 1.** Phylogenetic relationships of cassava NPF transporters.
DOI: https://doi.org/10.7554/eLife.19466.022

**Figure supplement 2.** Accumulation of 4MTB and I3M in *X. laevis* oocytes expressing close *M. esculenta* homologs of Me14G074000.
DOI: https://doi.org/10.7554/eLife.19466.023

**Figure supplement 3.** Expression analysis of YFP-tagged (C-terminal) GTR homologs from cassava in *X. laevis* oocytes.
DOI: https://doi.org/10.7554/eLife.19466.024

We believe that these genes provide a suitable model system for future studies that will investigate if the observed changes in transporter electrogenicities are caused by changes in coupling stoichiometry. This will lead to a mechanistic understanding of how substrate specificity and coupling stoichiometries co-evolve within the NPF family.

## A model for an evolutionary path of transporter substrate specificity

Based on our findings we propose a model for the evolutionary path of glucosinolate transporter substrate specificity in the NPF family (*Figure 7*). By tracking the evolution of GTR transporter specificity towards glucosinolates, we propose that a duplication event introduced permissive mutations in a high affinity, electrogenic transporter of the ancestral cyanogenic glucosides (represented here by MeCGTR1) to generate a 'transition' transporter with broad, non-electrogenic glucosinolate specificity and non-electrogenic cyanogenic glucoside specificity (represented by Me14G074000) (*Figure 7*). With the advent of glucosinolate biosynthesis and through further duplication and evolutionary divergence, our data suggests that the dual-specificity transporter lost its cyanogenic glucoside transport capacity and became a high affinity, electrogenic broad-specific glucosinolate transporter (represented by BrGTR1, AtGTRs, CpGTRL1 and CpGTRL2). The retainment of Me14G074000 in the cassava genome indicates that it may fulfill an important role in transport of cyanogenic glucosides despite its inferior transport properties compared to Me15G180400. Alternatively, its retainment could be explained by specificity towards other yet unidentified substrates. Presently, we can also not exclude that Me14G074000 represents an ancestral non-electrogenic, multi-specificity transporter, which through duplication and subfunctionalization gave rise to the electrogenic transporters with high affinity for cyanogenic glucosides (represented by MeCGTR1) and later for glucosinolates (represented by CpGTRL1, CpGTRL2, BrGTR1 and AtGTRs), respectively (*Figure 7*). Further subfunctionalization within the GTR clade led to the evolution of the GTR3 subclade identified as transporters with preference and high affinity for indole glucosinolate. Thus, the subfunctionalization within the GTRs from broad to narrow specificity is contrary to the evolutionary dynamics proposed previously for substrate-transport evolution, where progenitor transporters had

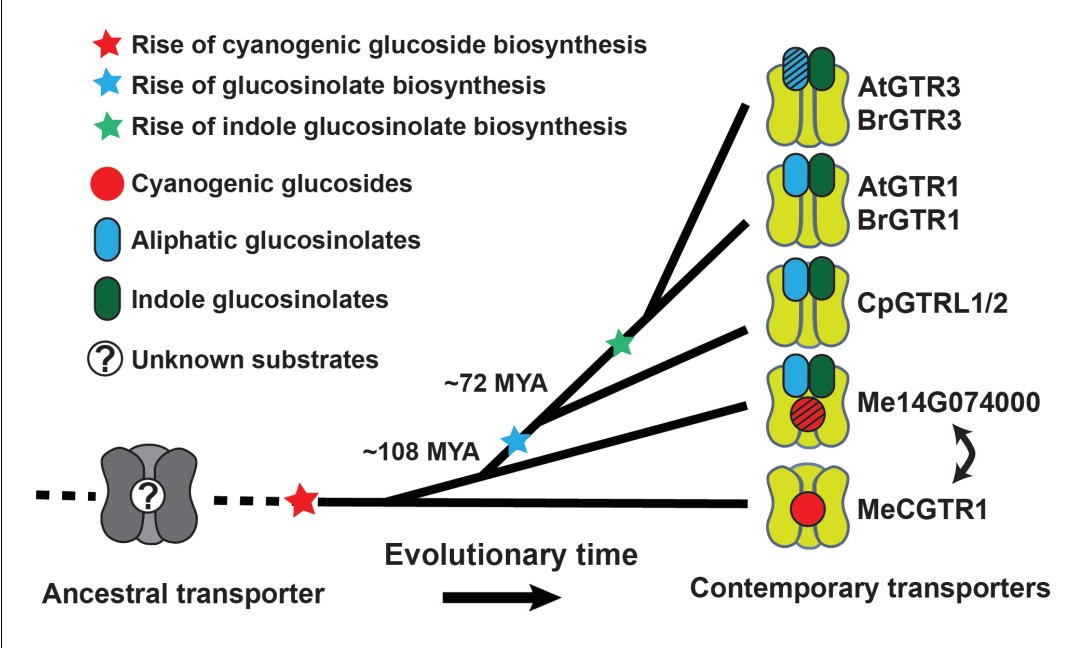

**Figure 7.** Model of the evolution of the glucosinolate NPF transporter specificity. We propose that diversification of an ancestral high-affinity cyanogenic glucoside transporter (exemplified by MeCGTR1) lead to a dual-specificity transporter capable of transporting both cyanogenic glucosides and glucosinolates (exemplified by Me14G074000). With the emergence of glucosinolate biosynthesis, high-affinity, broad-specific glucosinolate transporters evolved (exemplified by CpGTRL1/2 and At/BrGTR1), which then further specialized to preferentially transport indole glucosinolates when indole biosynthesis emerged. Bidirectional arrow indicates an alternative model where high-affinity cyanogenic glucoside transporters emerged from the dual-specificity transporter (exemplified by Me14g074000). *A. thaliana* and *C. papaya* or *M. esculenta* diversified 108 MYA (median, 26 studies) or 72.1 MYA (median, 8 studies), respectively (*Hedges et al., 2006*). Branch points represent likely duplication events that led to new transporter substrate specificities. Striped pattern indicates a transporter that is unable to over-accumulate substrate compared to external media.
DOI: https://doi.org/10.7554/eLife.19466.025

a narrow substrate specificity that expanded during evolution to become increasingly broad (*Lionarons et al., 2012*).

## Structural perspectives on glucosinolate and cyanogenic glucoside transporter substrate specificity

The large plant NPF family is homologous to the SLC15/PepT/PTR/POT families in bacteria and animals (*Léran et al., 2014*; *Daniel et al., 2006*). Several bacterial POTs (*Doki et al., 2013*; *Solcan et al., 2012*; *Newstead et al., 2011*) and one plant NPF homolog (AtNPF6.3) (*Parker and Newstead, 2014*; *Sun et al., 2014*) have been crystalized along with their substrates. Hence, it is possible to discuss the substrate specificities determined in the present study in a structural context by analysing the amino acid residues that are key for substrate interaction. We constructed an alignment comprising the sequence of AtNPF6.3, the crystalized bacterial POTs and the glucosinolate and cyanogenic glucoside transporters investigated in this study (*Figure 8—figure supplement 1*). From the structural studies on bacterial POTs and NPF6.3 (*Doki et al., 2013*; *Solcan et al., 2012*; *Parker and Newstead, 2014*; *Sun et al., 2014*; *Aduri et al., 2015*), the key substrate interacting amino acid residues were compiled, numbered P1-P13 (*Table 1*) and located in the alignment (*Figure 8—figure supplement 1*). In addition, we constructed homology models using the recent structure of NPF6.3 as a template and depicted amino acid positions P1-P13 within the models (*Figure 8* inserts). Analysis of AtGTR1, AtGTR3, Me14g074000 and MeCGTR1 homology models showed the P1-P13 residues to be exposed to the central substrate binding cavity of the transporters (*Figure 8*) and therefore to constitute candidates for substrate specificity determining residues. Five residues (P1-P3, P8 and P13) were conserved in all genes in the alignment. These constitute the EXXE[R/K] motif that has an indispensable role in proton coupling (*Table 1*) (*Solcan et al., 2012*; *Aduri et al., 2015*; *Jørgensen et al., 2015*). This suggests that the proton coupling mechanism is conserved

**Table 1.** Key peptide and nitrate substrate interacting amino acid residues from selected studies (*Doki et al., 2013; Solcan et al., 2012; Parker and Newstead, 2014; Sun et al., 2014; Aduri et al., 2015; Newstead et al., 2011; Lyons et al., 2014; Jørgensen et al., 2015*).

Corresponding residues in AtGTR1, AtGTR2, AtGTR3, CpGTRL1, CpGTRL2, Me14g74000 and MeCGTR1 were identified based on the alignment shown in *Figure 8—figure supplement 1*.

| Position | Function | AtGTR1 | AtGTR2 | AtGTR3 | CpGTRL1 | CpGTRL2 | Me14g74000 | MeCGTR1 | NPF6.3 | PepT1 | PepT$_{St}$ | PepT$_{GK}$ | PepT$_{SO}$ |
|---|---|---|---|---|---|---|---|---|---|---|---|---|---|
| P1 | EXXE[R/K] (*Doki et al., 2013; Aduri et al., 2015; Jørgensen et al., 2015*) | E75 | E57 | E33 | E30 | E49 | E54 | E38 | E41 | E23 | E22 | E32 | E21 |
| P2 | EXXE[R/K] (*Doki et al., 2013*) | E78 | E60 | E36 | E33 | E42 | E57 | E41 | E44 | E26 | E25 | E35 | E24 |
| P3 | EXXE[R/K] (*Doki et al., 2013*) | K79 | K61 | K37 | K34 | K43 | K58 | K42 | R45 | R27 | R26 | R36 | R25 |
| P4 | Peptide specificity (*Solcan et al., 2012*) | I82 | I64 | I40 | I37 | I46 | A61 | T45 | T48 | Y30 | Y29 | Y39 | F28 |
| P5 | Peptide specificity (*Doki et al., 2013*) | I83 | I65 | V41 | I38 | I47 | I62 | V46 | L49 | Y31 | Y30 | Y40 | Y29 |
| P6 | Peptide specificity (*Doki et al., 2013*) | L86 | L68 | S44 | L41 | L50 | L65 | S49 | G52 | R34 | R33 | R43 | R32 |
| P7 | Peptide binding (*Doki et al., 2013*) | N116 | N98 | N74 | N71 | N80 | N95 | N79 | F82 | Y64 | Y68 | Y78 | Y68 |
| P8 | Exxer-interactor (*Doki et al., 2013*) | R196 | R180 | R156 | R152 | R162 | R176 | R161 | K164 | K140 | K126 | K136 | K127 |
| P9 | Peptide binding (*Doki et al., 2013*) | T230 | T214 | T190 | T186 | T196 | T210 | T195 | N198 | N171 | N156 | N166 | N158 |
| P10 | Peptidebinding /protonation (*Parker and Newstead, 2014; Sun et al., 2014*) | I385 | I369 | Y343 | I341 | V351 | I365 | L350 | H356 | F297 | E299 | Q309 | F315 |
| P11 | Peptide specificity (*Doki et al., 2013*) | T386 | T370 | I344 | I342 | I352 | V366 | I351 | A357 | D298 | E300 | E310(Q) | D316 |
| P12 | Peptide binding (*Doki et al., 2013*) | L419 | L403 | L377 | L375 | L385 | V399 | T384 | Y388 | N329 | N328 | N342 | N344 |
| P13 | Proton translocation (*Doki et al., 2013*) | E513 | E497 | D471 | E469 | E479 | E493 | E478 | E476 | E595 | E400 | E413 | E419 |

DOI: https://doi.org/10.7554/eLife.19466.028

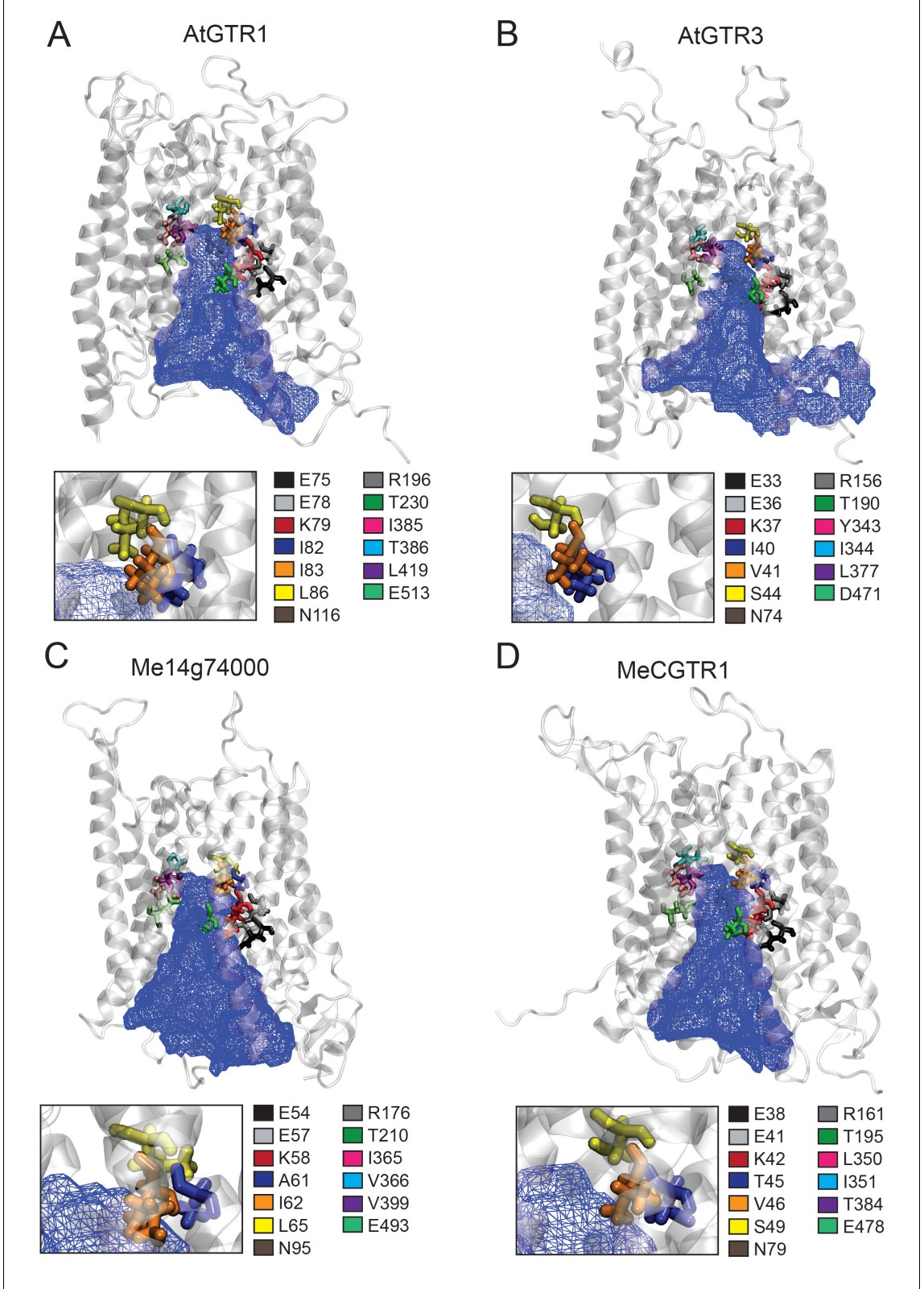

**Figure 8.** Putative substrate binding site of GTR1, GTR3, Me14g74000 and MeCGTR1. Homology modeling of GTR1, GTR3, Me14g74000 and MecGTR1 was carried out using NPF6.3 as template (see Materials and methods for details). Residues P1–13 are shown and color-coded according to legend. In blue mesh is the 3V determined central cavity (**Voss and Gerstein, 2010**). The inserts show P4, P5 and P6 (see text for discussion).
DOI: https://doi.org/10.7554/eLife.19466.026

*Figure 8 continued on next page*

*Figure 8 continued*

The following figure supplement is available for figure 8:

**Figure supplement 1.** Alignment of glucosinolate and cyanogenic glucoside transporters with NPF6.3 and selected POT transporters.
DOI: https://doi.org/10.7554/eLife.19466.027

regardless of the substrate specificity of a NPF transporter. Amino acid positions P7 and P9 are conserved respectively as asparagine and threonine across the GTR and cyanogenic glucoside transporters, whereas they are glycine and asparagine residues in POT transporters and phenylalanine and asparagine residues in NPF6.3. The only moiety shared between glucosinolates and cyanogenic glucosides is the glucose moiety. Thus, P7 and P9 could be involved in the interaction with the glucose moiety (*Table 1*, *Figure 1* and *Figure 6A*). Amino acid positions P10-P12 are not conserved across the GTRs, NPF6.3 and the POT transporters and the mutational pattern does not correlate to the changes we have seen in substrate specificity. Consequently, the role of these residues remains unclear. In contrast, amino acid positions P4, P5 and P6 (*Figure 8* inserts) show a conservation pattern that is consistent with the changes in substrate specificity shown in this study. This indicates that position P4, P5 and P6 may contribute to determining the substrate specificity of glucosinolate and cyanogenic glucoside transporters. Despite recent advances in understanding substrate specificity of peptide transporters (as outlined above), nothing is currently known about amino acid changes that determine transporter substrate specificity from an evolutionary perspective. Our work provides a framework for future studies to determine the amino acid changes that leads to substrate specificity changes during evolution of metabolite transporters.

## Conclusion and perspectives

With the dawning of cellular life, primitive membrane structures leading to today's complex phospholipid membranes necessitated membrane proteins to facilitate movement of structurally diverse compounds across cellular membranes (*Mansy et al., 2008*). Towards understanding the evolutionary paths that lead to new transporter substrate specificities, we show that before a substrate emerges, transporter specificity for the substrate may be present in transporters of chemically similar, more ancient substrates. As new substrates emerges (in this case glucosinolates), gene duplications allows such multifunctional transporters to diversify through subfunctionalization into transporters with greater specificity for the new substrate.

Thus, our findings support one model (*Bridgham et al., 2006*) to the problem posed by the classical evolution model – about how a new function (here transporter substrate specificity) can be selected for unless the substrate is there. We propose that redundant ancestral transporters created by gene duplication remain active due to a multifunctional specificity. When the new substrate emerges these ancestral transporters can be recruited and evolve into transporters with greater specificity for the new compound. Moreover, from a mechanistic perspective, our data suggests that the evolution of new substrate specificities in coupled secondary transporters is accompanied by changes in the electrogenic properties of the transport mechanism. Unraveling the structural determinants underpinning stoichiometry and binding of substrate constitutes a new frontier for understanding the birth and development of new transporter substrate specificities at the molecular level. This not only applies to the universal NPF transporters, including their drug-delivering mammalian counterparts (*Brandsch, 2013*), but for transporters in general. From an agro-biotech perspective, the identification of a cyanogenic glucoside transporter with high apparent affinity supports a prominent role for NPF transporters in specialized metabolism (*Nour-Eldin et al., 2012*; *Nour-Eldin and Halkier, 2013*) and opens new possibilities for controlling cyanogenic glucoside content in edible parts of crops such as bitter almond (*Dicenta et al., 2002*), barley (*Erb et al., 1979*) and cassava (*McMohan et al., 1995*) through transport engineering strategies (*Nour-Eldin and Halkier, 2013*).

## Materials and methods

### Gene names and IDs

The genes cloned and tested in this study are named as follows: (BrH02396), BrF01711, CpGTRL2 (Phytozome ID: evm.TU.supercontig_17.190), CpGTRL1 (Phytozome ID: evm.model.

supercontig_17.189), Cp17.188 (Phytozome ID: evm.model.supercontig_17.188), Tc1EG013796 (Phytozome ID: Thecc1EG013796), Me14G074000 (Phytozome ID: Manes.14G074000, cassava4.1_004026m), Me15G176100 (Phytozome ID: Manes.15G176100, cassava4.1_004213m), Me09G097200 (Phytozome ID: Manes.09G097200, cassava4.1_034015m), Me01G191900 (Phytozome ID: Manes.01G191900, cassava4.1_025742m), Me14G074100 (Phytozome ID: Manes.14G074100 cassava4.1_034466m), MeCGTR1 (Phytozome ID: Manes.15G180400, cassava4.1_004125m) and Me17G124600 (Phytozome ID: Manes.17G124600, cassava4.1_029616m).

## Cloning of synthesized genes into *Xenopus* expression vector

Design and direct assembly of synthesized uracil-containing non-clonal DNA fragments into vectors by USER cloning is described in more detail at Bio-protocol (*Jørgensen et al., 2017b*). All coding sequences were codon optimized for expression in *X. laevis* (NCBI Taxon: 8355) oocytes and synthesized as linear uracil containing DNA fragments (uStrings) by ThermoFisher Scientific Geneart. Each coding sequence was surrounded by the 8 bp USER tails that enable insertion into the USER compatible *X. laevis* expression vector pNB1u (*Geu-Flores et al., 2007*; *Nour-Eldin et al., 2006*). Each fragment contained a uracil at the appropriate location in each USER tail. The uracil was incorporated during synthesis. Thus, uStrings are mixed directly with the digested pNB1u vector without prior PCR amplification with uracil containing primers. Briefly, each uString was diluted to 100 ng/ul in $H_2O$. The USER-compatible pNB1u *X. laevis* oocyte expression vector was digested with *PacI/Nt. BbvCI* overnight, gel purified and diluted to a concentration of ~50 ng/ul (as previously described [*Nour-Eldin et al., 2006*; *MacAulay et al., 2001*]). For the USER reaction, 100 ng uStrings was mixed with 50 ng digested pNB1u, 1 unit USER enzyme (NEB-M5505S), 2 μl 5xPCR reaction buffer and 5 μl $H_2O$. The reaction was incubated at 37°C for 25 min, followed by 25 min at room temperature. The reaction mixture was used to transform chemically competent *E. coli* cells, plated on carbinicilin-containing LB plates. Selected colonies were grown overnight and extracted plasmids sequenced. All uStrings were inserted successfully into the pNB1u vector and out of the 13 genes synthesized and cloned, we had to sequence a second colony for only one of the genes. The fidelity and efficiency of cloning uStrings directly by USER cloning is satisfactory.

## Fluorophore tagging of transporters

For fluorophore tagging, coding sequences were PCR amplified from the expression constructs without the stop codon using uracil containing primers (see Materials and methods list of USER primers). The PCR fragments were USER cloned (as described previously [*Nour-Eldin et al., 2006*] into an oocyte expression vector (pNB1u variant, pLIFE22 [*Jørgensen et al., 2015*]) that translationally fuses the inserted coding sequence to a C-terminal YFP fluorophore, which is contained in the vector.

## List of USER primers for fluorophore-tagging of transporters. USER-overhang is in bold and a vector-specific double cysteine is underlined

| Primer | Name | Sequence |
|---|---|---|
| Forward | CpGTRL1 | **GGCTTAAU**ATGGAAAGGGCTGCCATGGC |
| Reverse no stop | CpGTRL1 | **GGTTTAAU**CCTCTGGACTCTTCGTTCACTTCG |
| Forward | Thecc1EG013796 | **GGCTTAAU**ATGGAAAAGAACGACAAAGAAGCC |
| Reverse no stop | Thecc1EG013796 | **GGTTTAAU**CCAACGAAGCTCTTGTCGCTCT |
| Forward | BrGTR3 | **GGCTTAAU**ATGGAAGTGGAAAAGACCCAGGAA |
| Reverse no stop | BrGTR3 | **GGTTTAAU**CCAACGGACACCTTGTCGAACTCG |
| Forward | Me15g176100 | **GGCTTAAU**ATGGAAGATAAGGAAGAGAAGTCC |
| Reverse no stop | Me15g176100 | **GGTTTAAU**CCCACAAGGTGTTTCTGAGACTGCTG |
| Forward | Me14g074100 | **GGCTTAAU**ATGGAAGTGGAACAGAGCGTGG |
| Reverse no stop | Me14g074100 | **GGTTTAAU**CCCTGCACCACTTCCAGAATCTTTGT |
| Forward | Me15g18400/MeCGTR1 | **GGCTTAAU**ATGGAAAACGGCAACGATCACG |

*Continued on next page*

*Continued*

| Primer | Name | Sequence |
|---|---|---|
| Reverse no stop | Me15g18400/MeCGTR1 | **GGTTTAAU**CCCACGTGGTGCTTCACGCTAG |
| Forward | Me17g124600 | **GGCTTAAU**ATGGAAAACAAAAAGCAGGAAACA |
| Reverse no stop | Me17g124600 | **GGTTTAAU**CCCAGGTCGCTTGGGATGAAAGAC |
| Forward | Me09g097200 | **GGCTTAAU**ATGGAAAACATGATTATCGCCAGC |
| Reverse no stop | Me09g097200 | **GGTTTAAU**CCGGCTGTAGCCTTCAGTTCCAGA |
| Forward | BrGTR1 | **GGCTTAAU**ATGGAAAGAAAGCCCTTCGAGGT |
| Reverse no stop | BrGTR1 | **GGTTTAAU**CCAACGCTGTTCTTAGCCTGCTT |
| Forward | Me14g074000 | **GGCTTAAU**ATGGCCACAGGCGAGACAATC |
| Reverse no stop | Me14g074000 | **GGTTTAAU**CCGGCCTTGATTGGCTTAACCTGC |
| Forward | CpGTRL2 | **GGCTTAAU**ATGGAAATGGACGGCAAAGAGC |
| Reverse no stop | CpGTRL2 | **GGTTTAAU**CCAACGTGGATGTTCTGCTTTTTCTT |
| Forward | Cp17.188 | **GGCTTAAU**ATGGCCTTCCTGCTGACCG |
| Reverse no stop | Cp17.188 | **GGTTTAAU**CCGATATCGCTCTGCTTGGTGCT |
| Forward | AtGTR1 | **GGCTTAAU**ATGAAGAGCAGAGTCATT |
| Reverse no stop | AtGTR1 | **GGTTTAAU**CCGACAGAGTTCTTGTC |
| Forward | AtGTR3 | **GGCTTAAU**ATGGAGGTTGAGAAGACAGAGAAG |
| Reverse no stop | AtGTR3 | **GGTTTAAU**CCCACTGACACCTTATCAAACTCAGC |

## Oocyte bioimaging

Oocyte bioimaging was performed essentially as previously described (*Geiger et al., 2011*), with the addition that oocytes expressing YFP-tagged transporters were mounted on a glass slide and a Kulori (90 mM NaCl, 1 mM KCl, 1 mM $MgCl_2$, 10 mM MES adjusted to pH7.4) solution with 20 µM FM4-64fx was added 1 min prior to bioimaging by confocal scanning microscopy using a SPX5-X Point-scanning Confocal from Leica Microsystems.

## Oocyte preparation and cRNA injection

*X. laevis* oocytes (stages V-VI) were purchased as defolliculated oocytes (stages V-VI) from Ecocyte Biosciences (Germany). Injection of 50 nl cRNA (500 ng/µl) into *X. laevis* oocytes was done using a Drummond NANOJECT II (Drummond scientific company, Bromall Pennsylvania). Oocytes were incubated for 3 days at 17°C in Kulori (90 mM NaCl, 1 mM KCl, 1 mM $MgCl_2$, 10 mM MES) pH7.4 prior to assaying.

## Phylogenetic analysis - Dataset assembly

NPF homologs (also called SLC15/PepT/PTR/POT [*Léran et al., 2014*; *Daniel et al., 2006*]) from *Arabidopsis thaliana(At)*, *Brassica rapa (Br)*, *Carica papaya (Cp)*, *Theobroma cacao (Tc)*, *Manihot esculenta (Me)*, *Glycine max (Gm)*, *Gossypium raimondii (Gr)*, *Medicago truncatula (Mt)* and *Solanum lycopersicum (Sl)* were retrieved from phytozome (http://www.phytozome.net) by searching for sequences classified as oligopeptide transporters (PFAM:PF00854 and Panther: PTHR11654). To remove pseudogenes, we predicted the number of transmembrane helices using TMHMM server V. 2.0 (*Sonnhammer et al., 1998*) and removed sequences with <6 transmembrane helices and fewer than 300 amino acids. Genes were renamed according to the following guidelines. evm.model. supercontig_139.55 from *C. papaya*, was renamed to 'Cp139.55'. Solyc10g024490.1 from *S. lycopersicum* was renamed to Sl10g024490.1. Thecc1EG035998 from *T. cacao* was renamed to Tc1EG035998. Glyma.18G097800 from *G. max* was renamed to Gm18G097800. Brara.C02073 from *B. rapa* was renamed to BrC02073. Medtr4g107510 from *M. truncatula* was renamed to Mt4g107510. Manes.16G072300 from *M. esculenta* was renamed to Me16G072300. Gorai.012G121800 from *G. raimondii* was renamed to Gr012G121800.

## Phylogenetic analysis - Alignment and phylogenetic analysis

Sequences were aligned using MUSCLE (*Edgar, 2004*) with a gap open penalty of −2.9, gap extend of 0 and hydrophobicity multiplier of 1.2. Poorly aligned regions were trimmed manually. Prottest v3.4.2 (*Darriba et al., 2011*) was used with final multiple sequence alignments to identify the appropriate LG-based phylogenetic models to use for subsequent work. The best fit (for phylogenies in *Figure 1A*, *Figure 1—figure supplement 1* and *Figure 6—figure supplement 2*) was LG+I+G+F that use a general amino acid replacement matrix (*Le and Gascuel, 2008*) with a proportion of invariable sites (+I) (*Reeves, 1992*), a gamma distribution for modelling the rate heterogeneity (+G) (*Yang, 1993*), and empirical amino acid frequencies (+F) (*Cao et al., 1994*). The best fit (for phylogenies in *Figure 5A*, *Figure 6B* and *Figure 5—figure supplement 1*) was LG+G+F. Bayesian inference trees were calculated using MrBayes 3.2.6 (*Huelsenbeck and Ronquist, 2001*) until convergence was reached ('average standard deviation of split frequencies' <0.01). The temperature heating parameter was set to 0.05 (temp = 0.05) to increase the chain swap acceptance rates, thereby reducing the chances of Markov chains to get stuck at local high-probability peaks. Burn-in was set to 25% (burninfrac = 0.25) and the number of Markov chains was set to 8 (nchains = 8). Maximum likelihood trees were produced with RAxML 8.2.3 using the LG PROTGAMMA model and 500 bootstrap replicates (*Stamatakis, 2014*). RAxML bootstrap values were portrayed on the MrBayes generated consensus tree. NRT1.1 from *Chlamydomonas reinhardtii* (Phytozome ID: Cre04.g224700) was used as the out-group. All analyses were run in MPI via the CIPRES SCIENCE GATEWAY (*Miller et al., 2011*) at the San Diego Supercomputer Center (SDSC). Trees were visualized in figtree (http://tree.bio.ed.ac.uk/software/figtree/) and annotated with Adobe Illustrator.

## Definition of biological versus technical replicates

'Biological replicates' denote replicated measurements using different biological cases, whereas 'technical replicates' use the same biological cases.

## Estimation of sample size

Through pilot experiments average variation between biological replicates have been determined. Sample sizes for this study were decided upon as the best compromise between average power and experimental constraints.

## Glucosinolate and cyanogenic glucoside uptake assays

Uptake assays in *Xenopus laevis* oocytes using liquid chromatography–mass spectrometry to detect transport activity is described in more detail at Bio-protocol (*Jørgensen et al., 2017a*). *X. laevis* uptake assays were carried out as follows: Oocytes were preincubated in Kulori pH 5 for 5 min, transferred to Kulori pH five with substrate for 60 min incubation, followed by four washes and transferred to Eppendorf tubes (one oocyte per tube). Excess washing buffer was removed and oocytes were busted in 50 µl of 50% MeOH (with sinigrin as internal standard) and the homogenate was left in the freezer for 2 hr to precipitate proteins. This was followed by centrifugation at 20,000 x g for 15 min to pellet remaining proteins. The supernatant was transferred to new tubes and diluted with 60 µl H2O. The diluted samples were filtered through a 0.45 µm PVDF based filter plate (MSHVN4550, Merck Millipore) and subsequently analyzed by analytical Liquid Chromatography – Mass Spectrometry. 4-methylsulfinylbutyl glucosinolate (4MTB) and 3-indolylmethylglucosinolate (I3M) were obtained from C$_2$ Bioengineering (http://www.glucosinolates.com/) and CFM Oskar Tropitzsch GmbH, Marktredwitz (http://www.cfmot.de/), respectively. Cyanogenic glucoside prunasin was synthesized by MSM as previously described (*Møller et al., 2016*). Cyanogenic glucoside linamarin was purchased from Santa cruz biotechnology.

## Desulfo glucosinolate analysis of *X. laevis* oocytes by LC-MS

ESI-LC-MS analysis of desulfo glucosinolates from *X. laevis* uptake assays were performed as described before (*Nour-Eldin et al., 2012*).

## Intact glucosinolate analysis of *X. laevis* oocytes by LC-MS

Extracts from uptake assays (see above) were directly analyzed by LC-MS/MS. Chromatography was performed on an Advance UHPLC system (Bruker, Bremen, Germany). Separation was achieved on a

Kinetex 1.7u XB-C18 column (100 × 2.1 mm, 1.7 µm, 100 Å, Phenomenex, Torrance, CA, USA). Formic acid (0.05%) in water and acetonitrile (supplied with 0.05% formic acid) were employed as mobile phases A and B respectively. The elution profile was: 0–0.2 min, 2% B; 0.2–1.8 min, 2–30% B; 1.8–2.5 min 30–100% B, 2.5–2.8 min 100% B; 2.8–2.9 min 100–2% B and 2.9–4.0 min 2% B. The mobile phase flow rate was 400 µl min$^{-1}$.The column temperature was maintained at 40°C. The liquid chromatography was coupled to an EVOQ EliteTripleQuad mass spectrometer (Bruker, Bremen, Germany) equipped with an electrospray ion source (ESI) operated in combined positive and negative ionization mode. The instrument parameters were optimized by infusion experiments with pure standards. The ion spray voltage was maintained at +5000 V or −4000 V for cyanogenic glucoside and glucosinolate analysis, respectively. Cone temperature was set to 300°C and cone gas to 20 psi. Heated probe temperature was set to 180°C and probe gas flow to 50 psi. Nebulizing gas was set to 60 psi and collision gas to 1.6 mTorr. Nitrogen was used as probe and nebulizing gas and argon as collision gas. Active exhaust was constantly on. Multiple reaction monitoring (MRM) was used to monitor analyte parent ion → product ion transitions: MRM transitions were chosen based on direct infusion experiments. Detailed values for mass transitions is found in Materials and methods list of primers.

MRM transitions for intact glucosinolates and cyanogenic glucosides determined by LC-MS/MS $^Q$Quantifier ion used for quantification of the compounds. Additional MRM transitions were used for compound identification. IS = internal Standard.

| Compound | Q1 | Q3 | CE [eV] | Internal standard | Response factor |
|---|---|---|---|---|---|
| SIN (Sinigrin. 2-propenyl-GLS) = IS | 358.0 | 97.0$^Q$ | 22 | n.a. | n.a. |
| | 358.0 | 75.0 | 30 | | |
| | 358.0 | 259.0 | 20 | | |
| 4MTB | 420.0 | 97.0$^Q$ | 23 | Sinigrin | 0.99 |
| | 420.0 | 75.0 | 30 | | |
| | 420.0 | 259.0 | 23 | | |
| I3M | 447.1 | 97.0$^Q$ | 10 | Sinigrin | 13.34 |
| | 447.1 | 259.0 | 10 | | |
| | 447.1 | 205.0 | 10 | | |
| Linamarin | 248.2 | 85.2$^Q$ | −15 | Sinigrin | 163.9 |
| | 248.2 | 97.3 | 19 | | |
| | 248.2 | 163.1 | −5 | | |
| Prunasin | 296.1 | 163.1$^Q$ | −4 | Sinigrin | 97.7 |
| | 296.1 | 85.2 | −17 | | |
| | 296.1 | 97.3 | −21 | | |

Both Q1 and Q3 quadrupoles were maintained at unit resolution. Bruker MS Workstation software (Version 8.2, Bruker, Bremen, Germany) was used for data acquisition and processing.

Linearity in ionization efficiencies was verified by analyzing dilution series of standard mixtures. Quantification of all compounds was achieved by use of sinigrin as internal standard.

## Calculating up concentration of substrate inside oocytes

The concentration of imported substrate was calculated based on previous reports determining the water content of oocytes to be ~70% of total volume (*de Laat et al., 1974*) and an oocyte diameter of 1.5 mm. Assuming this, the oocyte cytosolic volume was estimated to be 1 µl allowing us to calculate the up-concentration of substrate.

## Nitrate uptake assays and analysis by ICPMS

Nitrate uptake assays were carried out as follows: Oocytes were preincubated in Kulori pH 5 for 5 min, transferred to Kulori pH 5 with $^{15}$N-labelled KNO$_3$ (Sigma Aldrich, 335134) at the indicated concentration for 60 min. Subsequently, oocytes were washed 4 times in H$_2$O and transferred to tin

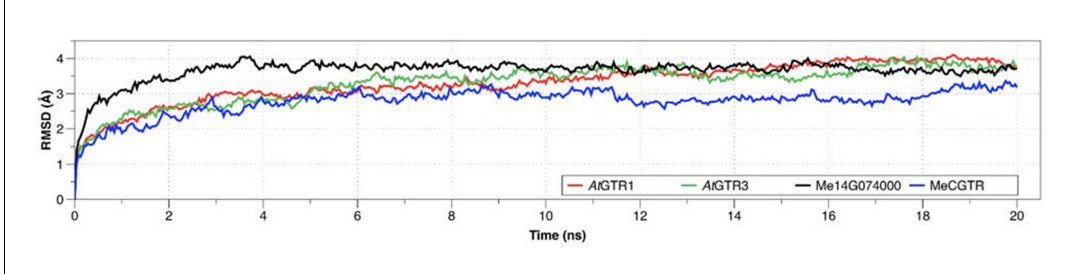

**Figure 9.** Root-mean square deviations (RMSD) of the position for all backbone atoms of the models 579 from their initial configuration as a function of simulation time.
DOI: https://doi.org/10.7554/eLife.19466.029

capsules prior to stable isotope analysis by IRMS (Isotope Ratio Mass Spectrometry). Stable isotope ratio analysis of nitrogen were conducted as described by Laursen et al. (2013). In brief, analysis were conducted using a Europa Scientific ANCA-SL Elemental Analyser coupled to a Europa Scientific 20–20 Tracermass mass spectrometer (Sercon Ltd., Crewe, UK). Quality control (accuracy and precision) was performed by analysis of standards and certified reference materials from the International Atomic Energy Agency, IAEA, Vienna, Austria and Iso-Analytical Limited, IA, Crewe, UK.

## Electrophysiological measurements

All measurements were performed with a Two Electrode Voltage-Clamp system (TEVC) composed of an NPI TEC-03X amplifier (NPI electronic GmbH, Germany) connected to a PC with pCLAMP10 software (Molecular devices, USA) via an Axon Digidata 1440a digitizer (Molecular devices, USA). Oocytes were placed in the recording chamber and perfused with a standard Kulori-based solution (90 mM NaCl, 1 mM KCl, 1 mM $CaCl_2$, 1 mM $MgCl_2$, 1 mM $LaCl_3$ and 10 mM MES pH 5). TEVC data was analysed in excel after extraction from pCLAMP10 software as a Microsoft Excel compatible worksheet. Substrate-dependent currents were calculated by subtracting currents before addition of substrate from currents after addition of substrate. Visualization and curve fitting to the Michaelis-Menten equation (*Equation 1*) to calculate the apparent $K_m$ value was done using SigmaPlot version 12.5/13.0 (Systat software, USA).

*Equation 1* - Michaelis-Menten equation. I is the current, $I_{max}$ is the maximal current achieved by the transporter at saturating concentrations of substrate.

$$I = \frac{I_{max} * [substrate]}{[substrate] + K_m} \tag{1}$$

SigmaPlot version 12.3 (Systat software, USA) was used for statistical analysis and data plotting.

## Plant materials and growth conditions

*Arabidopsis thaliana* ecotype Columbia-0 (Col-0) (NCBI Taxon: 3702) lines and three insertion mutants gtr1 (SAIL_801_G03), *gtr2* (SAIL_20_B07) and *nrt1.9–2/gtr3* (GK-099B01) were obtained from NASC. To construct double and triple mutants, *gtr3* homozygous were crossed to a previously characterized and published gtr1 gtr2 dko (*Nour-Eldin et al., 2012*). F1 progeny of those crosses were all phenotypically normal. The resulting F2 progeny (160 plants) were screened by PCR, and homozygous mutants, (gtr1 gtr3, gtr2, gtr3) were obtained. Seeds from self-pollination of gtr1+/-gtr2 gtr3 plants were collected and allowed to self-pollinate. The following F3 progeny were screened by PCR and homozygous gtr1 gtr2 gtr3 mutants were identified (*Figure 4—figure supplement 3*).

## Rosette and root analysis of *A. thaliana* plants

Plants used for rosette and root analysis were grown from sterilized *A. thaliana* seeds put onto 0.5 mL PCR tubes, which had been filled with agar (1% (w/v) sucrose) and cut at the bottom. A total of 48 tubes were placed into a yellow pipette tip box filled with nutrient solution (one-half strength Murashige and Skoog basal medium). Afterwards, the box was sealed with and incubated in a

growth chamber under cultivated at 12 hr days with 70% relative humidity, and a light intensity of 100 mE. Upon root emergence into the growth media, PCR tubes containing the seedlings were transferred on to a perforated screw cap of 50 mL Falcon tubes. The cap was screwed onto the tube and the tubebottom was cut off to allow the root to grow into the media. A total of 30 bottles were arrayed in a closed plastic box filled with MS media. Air was pumped into the media by four tubes from holes at four corners of the box. The box was put into the same growth chamber and plant material was harvested for glucosinolate analysis after two weeks. For micro-grafting, seeds were surface-sterilized by washing in 70% (v/v) ethanol containing 0.05% (v/v) Triton X100 for 5 min followed by washing in water, left on sterile filter paper, and sown on half-strength MS agar plates. The plates were cold-stratified for 2 days followed by vertical growth for 3 to 5 d under long-day conditions (light: 16 hr, 20°C; darkness: 8 hr, 16°C).

## Micro-grafting of *A. thaliana* seedlings

Micro-grafting of *A. thaliana* seedlings was performed in a laminar flow cabinet using a dissection microscope as described (*Andersen et al., 2014*). Briefly, Arabidopsis seedlings grown on MS-containing agar (without sugar) for 4 days were transferred to a sterile one layer thick wet nitrocellulose filter (Whatman NC 45 ST) and two layers of filter paper in a sterile petri dish. The cotyledons were removed and incisions were made on the hypocotyl close to the shoot using a sapphire knife. The root stocks and scions of seedlings were joined using sterile forceps. The plates were sealed using Micropore tape (3M) and incubated vertically under long-day conditions (light: 16 hr, 20°C; darkness: 8 hr, 16°C). Successfully joined seedlings were transferred to MS agar and kept under long-day conditions until the age of 3 weeks and analyzed by LC-MS.

## Desulfo glucosinolate analysis of plant material by LC-MS

Glucosinolates were analyzed as desulfo-glucosinolates by UHPLC/TripleQuad-MS. Chromatography was performed on an Advance UHPLCTM system (Bruker, Bremen, Germany) equipped with a C-18 reversed phase column (Kinetex 1.7 u XB-C18, 10 cm x 2.1 mm, 1.7 µm particle size, Phenomenex, Torrance, CA, USA) by using a 0.05% formic acid in water (v/v) (solvent A)−0.05% formic acid in acetonitrile (v/v) (solvent B) gradient at a flow rate of 0.4 ml*min−1. The column temperature was maintained at 40°C. The gradient applied was as follows: 2% B (0.5 min), 2–30% (0.7 min), 30–100% (0.8 min), 100% B (0.5 min), 100–2% B (0.1 min), and 2% B (1.4 min). The liquid chromatography was coupled to an EVOQ Elite TripleQuad mass spectrometer (Bruker, Bremen, Germany) equipped with an electrospray ion source (ESI) operated in positive ionization mode. The ion spray voltage was maintained at +3500 V. Cone temperature was set to 300°C and cone gas to 20 psi (arbitrary units). Heated probe temperature was set to 400°C and probe gas flow set to 40 psi. Nebulizing gas was set to 60 psi and collision gas to 1.6 mTorr. Desulfo-glucosinolates were monitored based on the following Multiple reaction monitoring (MRM) analyte parent ion → product ion transitions [Collision energy]: 3-methylthiopropyl (3mtp, m/z 328 → 166 [5V]); 3-methylsulfinyl (3msp, m/z 344 → 182 [10V]); 2-propenyl (2-prop, m/z 280→ 118 [5V]); 3-hydroxypropyl (3ohp, m/z 298 → 118 [15V]); 3-benzoyloxy (3bzo, m/z 402 collision gas to 1.6 mTorr. Desulfo-glucosinolates were monitored based on the following Multiple reaction monitoring (MRM) analym/z 294 → 132 [15V]); (R/S)−2-hydroxy-3-butenyl, m/z 310 → 130 [15V]; 4-hydroxybutyl (4ohb, m/z 312 ision gas to 1.6 mTorr. Desulfo-glucosinolates were monitored based on the following Multiple reaction monitoring (MRM) analym/z 294 → 132 [15V]); (R/S)−2-h-methylsulfinylheptyl (7msh, m/z 400 → 238 [7V]); 8-methylthiooctyl (8mto, m/z 398 → 236 [5V]); 8-methylsulfinyloctyl (8mso, m/z 414 → 252 [5V]); indol-3-ylmethyl (I3M, m/z 369 → 207 [10V]); N-methoxy-indol-3-ylmethyl (NMOI3M, m/z 399 → 237 [10V]); 4-methoxy-indol-3-ylmethyl (4MOI3M, m/z 399 → 237 [10V]); p-hydroxybenzyl (pOHB, m/z 346 346 6 OHB, m/z 346 r. Desulfo-glucosinolates were monitored based on the following Multiple reaction monitoring (MRM) relative to the internal standard pOHB calculated from standard curves in control extracts.

## Transporter homology modelling

Homology models for AtGTR1, AtGTR3, Me14G074000 and MeCGTR1 was build using NPF6.3 (PDB: 4OH3) as template (*Sun et al., 2014*). Transporter homology models were built and optimized using Prime (*Jacobson et al., 2004*) included in the Schrödinger suite (www.schrodinger.com). All homology models were validated using PROCHECK (*Laskowski et al., 1993*).

Homology models were embedded into a pre-equilibrated phosphatidyl oleoyl phosphatidylcholine (POPC) bilayer in a periodic boundary condition box with pre-equilibrated Simple Point Charge (SPC) water molecules in addition to $Na^+$ and $Cl^-$ ions corresponding to a 150 mM buffer. Each system was subjected to a conjugate gradient energy minimization and relaxed by short molecular dynamics simulations (MDs) using the default 'Relax model system' protocol implemented in Desmond (*Bowers et al., 2006*) followed by 20 ns of MDs with periodic boundary conditions. A restriction was applied to the secondary structure of the transporters using a spring constant force of 0.5 kcal $\times$ mol$^{-1}$ $\times$ Å$^{-2}$. The simulation temperature was set to 300K, and both temperature and pressure were kept constant during the MDs (NPT ensemble simulation) using the Nose-Hoover chain thermostat method (*Martyna et al., 1992*) and the Martyna-Tobias-Klein barostat method (*Martyna et al., 1994*). Coordinates were stored every 2 fs. The MDs were run on a GPU computing cluster at the University of Talca, Chile, using 1 GPU GeForce GTX 980 for each simulation. The root-mean square deviations (RMSD) of the position for all backbone atoms of the models from their initial configuration as a function of simulation time are illustrated *Figure 9*. All models were equilibrated after 4 ns of MDs (except AtGTR3, which reached equilibrium around 7 ns). The RMSD values remain within 4 Å for all models, demonstrating the conformational stability of the transporter structures.

To determine the intracellular cavity/channel we used 3V (*Voss and Gerstein, 2010*) via the web interface found at http://3vee.molmovdb.org.

VMD (*Humphrey et al., 1996*) was used for visualizing and displaying homology models and cavity/channel.

## Acknowledgements

Prof. Michael Broberg Palmgren and Prof. Daniel J Kliebenstein are thanked for help with phylogenetic analyses, critical reading and constructive comments on the manuscript. Technician Anja Hecht Ivø and Assist. Prof. Kristian Holst Laursen are thanked for technical assistance with the IRMS analyses. Imaging data were collected at the Center for Advanced Bioimaging (CAB) Denmark, University of Copenhagen. MEJ, DX, CC, HHN and BAH were funded by DNRF99 grant from the Danish National Research Foundation.

## Additional information

### Competing interests

Morten Egevang Jørgensen, Hussam Hassan Nour-Eldin, Barbara Ann Halkier: Authors declare a competing financial interest, as transport engineering of glucosinolates is the subject of a patent (WO2012004013-A2 and WO2012004013-A3) published January 12th 2012. The other authors declare that no competing interests exist.

### Funding

| Funder | Grant reference number | Author |
| --- | --- | --- |
| Danish National Research Foundation | DNRF 99 | Morten Egevang Jørgensen<br>Deyang Xu<br>Christoph Crocoll<br>Mohammed Saddik Motawia<br>Carl Erik Olsen |

The funders had no role in study design, data collection and interpretation, or the decision to submit the work for publication.

### Author contributions

Morten Egevang Jørgensen, Identified GTR3 as an indole-specific glucosinolate transporter, Designed the study, Performed the phylogenetic analyses, Biophysical characterization, Data analysis and contributed to generating the gtr1 gtr2 gtr3 triple mutant and micrografting, Wrote the paper based on a draft; Deyang Xu, Generated the gtr1 gtr2 gtr3 triple mutant, Performed plant work

including micro-grafting and glucosinolate analyses of transporter mutants and contributed to study design and preparation of the manuscript; Christoph Crocoll, Carl Erik Olsen, Performed LC-MS analyses; Heidi Asschenfeldt Ernst, David Ramírez, Osman Mirza, Performed multiple sequence alignments, homology modelling; Mohammed Saddik Motawia, Synthesized prunasin; Hussam Hassan Nour-Eldin, Performed and analysed nitrate uptake assays, Advised and contributed to the study design, Wrote the paper based on a draft, All authors discussed the results and commented on the manuscript; Barbara Ann Halkier, Advised and contributed to the study design, Wrote the paper based on a draft, All authors discussed the results and commented on the manuscript

### Author ORCIDs
Morten Egevang Jørgensen ⓘ http://orcid.org/0000-0001-6503-0495
Deyang Xu ⓘ http://orcid.org/0000-0003-1015-6458
Hussam Hassan Nour-Eldin ⓘ http://orcid.org/0000-0001-6660-0509
Barbara Ann Halkier ⓘ http://orcid.org/0000-0002-3742-702X

### Decision letter and Author response
Decision letter https://doi.org/10.7554/eLife.19466.033
Author response https://doi.org/10.7554/eLife.19466.034

## Additional files

### Data availability
The following previously published dataset was used:

| Author(s) | Year | Dataset title | Dataset URL | Database and Identifier |
|---|---|---|---|---|
| Mustroph, Angelika Zanetti, M Eugenia Jang, Charles J H Holtan, Hans E Repetti, Peter P Galbraith, David W Girke, Thomas Bailey-Serres, Julia | 2009 | Profiling translatomes of discrete cell populations resolves altered cellular priorities during hypoxia in Arabidopsis | https://www.ncbi.nlm.nih.gov/geo/query/acc.cgi?acc=GSE14493 | Publicly available at NCBI Gene Expression Omnibus (accession no: GSE14493), 10.1073/pnas.0906131106 |

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
