## [Decision Letter]

Thank you for submitting your article "Origin and evolution of a transporter substrate specificity" for consideration by *eLife*. Your article has been reviewed by three peer reviewers, one of whom is a member of our Board of Reviewing Editors, and the evaluation has been overseen by Richard Aldrich as the Senior Editor. The following individual involved in review of your submission has agreed to reveal his identity: Markus Geisler (Reviewer #3).

The reviewers have discussed the reviews with one another and the Reviewing Editor has drafted this decision to help you prepare a revised submission.

In the manuscript, the authors explore potential evolution of substrate specificity in glucosinolate transporters (belonging to SLC15 family) in relation to the evolution of the corresponding metabolic pathways. They found that the evolution of the specificity for glucosinolates from earlier transporters of cyanogenic glucosides proceeded through a broad specificity transporter, capable of transporting both type of substrates. Once glucosinoalte specificity emerged, some of the family members became more narrowly specialized for specific glucosinolates. Interestingly, during evolutionary transition of the specificity from cyanogenic glucosides to glucosinolates, the broad-specificity transporter showed non-electrogenic transport of cyanogenic glucosides, which contrasts with both earlier and later members, which all showed proton-driven transport.

The reviewers found the work interesting, original and of considerable potential impact. High quality of experimental work was pointed out. Unfortunately, a number of significant issues have been identified that seem to be essential to for the conclusions that the authors draw.

Key issues:

1) The key point of the paper seems to be that Me14G074000 is the transition stage between transporters specific for cyanogenic gluocosides and glucosinlates. Could it be that Me14G074000 has an entirely different "native" substrate? It is odd that in a family of H-coupled transporters, there would be one that takes up relevant substrate, cyanogenic glucoside in a non-coupled manner. The authors suggest that cassava contains another transporter that would be specific for cyanogenic glucosides, while the Me14G074000 is a "transition" transporter. If so, a closely related specific transporter should be easily identified. The authors should attempt to find it.

2) At least negative (preferably all) transport data need to be confirmed by expression analyses, such as a simple western and/or confocal analyses. This is in the transport field standard as absence of transport can have many reasons (such as no expression or miss-targeting etc.), while high transport rates can be simply caused by higher expression levels. For example, an important conclusion that the transporter from cacao does not transport glucosinolates is based on simple lack of activity, which is not convincing in the absence of expression analysis or a positive control, showing that this transporter transports its native substrate.

3) The study includes phylogenetic analysis but omits any detail about specific residue changes between these transporters that could and should account for their biochemical differences. Given the recent structure of the NRT1.1 homologue and bacterial POT family members, the authors should examine the amino acid sequence changes between their characterized homologues and put these changes into structural perspective. They should discuss their findings in light of known conserved motifs.

4) The authors should interpret their results in structural perspective, perhaps mapping the amino acid differences on a structural model of the transporter and discuss potential substrate-binding site. We point out that in the NRT1.1 structural paper some of the key amino acids differing between NRT1.1 and distantly related bacterial peptide transporters were identified. Perhaps, these would be helpful in examining differences between GRTs and closely related transporters.

5) The proton coupling mechanism of GTR3 in uptake of 4MTB is unclear. From the apparent lack of currents it can be concluded that the uptake is coupled to 1 proton, but the substrate concentrations used in this experiment (Figure 2A) are very low. How will the currents look like at higher 4MTB concentrations, where the uptake is more significant? Substrate-dependent currents should be examined and coupling mechanism should be established as it is very surprising that the mechanism seems to be 1:1 yet there is no accumulation in Figure 2D. Could the reversal potentials be used to better establish the coupling stoichiometry in these and other transporters examined in the study?

6) The finding that GTRL1/2 from Carica papaya transports both 4MTB and I3M is surprising and used as argument for the concept that glucosinolate transporter specificity arose from broad to narrow. Also, analysis of Me14G074000 from Cassava that transport both 4MTB and I3M supports this assumption. While we understand why these organisms have been chosen, we feel hat their individual choice and the number of tested transporters and species (here only 1 each) is too small to fully support these very important conclusions. We suggest including at least a second transporter from a similar but distant organism.

7) A broader literature review is needed to discuss what is known in the field of evolution regarding development of novel specificities. This has previously been examined for hormone receptors, for example, in great detail.

[Editors' note: further revisions were requested prior to acceptance, as described below.]

Thank you for resubmitting your work entitled "Origin and evolution of transporter substrate specificity within the NPF family" for further consideration at *eLife*. Your revised article has been favorably evaluated by Richard Aldrich (Senior editor), a Reviewing editor, and one reviewer.

The manuscript has been improved but there are some remaining issues that need to be addressed before acceptance, as outlined below:

After additional discussion, we feel that the authors have not sufficiently addressed the key issue 2) of the original review. We would argue that based on the data presented, it is not possible to make definitive statements about stoichiometry of transport. Specifically, we think that the presence of concentrative transport is clearly indicative of ion-coupled transport. Such concentrative transport is demonstrated for the uptake of glucosinolates by Me14G0176100 (Figure 5B) as well as by GTR1s and GTR3s (I3M). In contrast, transport of cyanogenic glucosides by Me14G0176100 is not concentrative (Figure 6 A and C). Thus, it is reasonable to hypothesize that it is not ion-coupled. However, the presence or absence of current is difficult to interpret based on the reported relatively preliminary electrophysiology experiments. For example, the observed current might be due to an uncoupled conductance that accompanies coupled transport and not due to stoichiometry of >2H^+^ per substrate. In that regard, we note that GTR1s and Me14G0176100 accumulate glucosinolates to a similar extent. Would not one expect GTR1s to concentrate substrates to a greater extent than Me14G0176100 if they operated with higher proton: substrate stoichiometry? Furthermore, the absence of current might be simply a consequence of slow rate of transport that would not allow detection of the ion flux. More detailed studies would need to be conducted to resolve these questions. We recognize that these go beyond the scope of the current manuscript. We urge the authors to re-evaluate their conclusions on the coupling mechanisms and to re-formulate them in more speculative, hypothetical terms.

Reviewer #2:

It seems to me the authors have not actually addressed key issue 1. This does weaken the strength of the implications made about the significance of the study and retention of Me14g74000 as a transition transporter.

Key issue 2. My hunch is that Me14g74000 is proton coupled, as it contains all the necessary residues identified in POT family transporters – namely P1, 2, 3 and P13 as described here. It is certainly possible that a change in substrate specificity would be accompanied by a change in stoichiometry. It would have been satisfying if further experimental evidence was provided to support this hypothesis, as this could be an important factor in the evolution of the binding site and transporters. However, I can see that this would increase the remit of the study beyond identification.

The inclusion of the structural comparison definately improves the manuscript and now allows the reader to appreciate the changes made to the binding site that differentiate the cyanogenic glucosides from the glucosinolates. It is frustrating that these cannot be narrowed further, but this is, again, likely beyond the scope of the present study. Incidently, P13 is likely not involved in peptide specificity but in proton translocation, as it is also essential for NRT1.1 function. The structural figures are perhaps a little confusing – I would have opted for an overlay.

Overall the manuscript is much improved, although I am not sure it provides certainty as to the ultimate conclusions drawn on the evolution of the NPF family. I have no major objections and think the study does explore an interesting aspect of transporter evolution.

[Editors' note: further revisions were requested prior to acceptance, as described below.]

Thank you for re-submitting your article "Origin and evolution of transporter substrate specificity within the NPF family" for consideration by *eLife*. Your revised article has been reviewed by the Reviewing Editor and Richard Aldrich as the Senior Editor.

The Reviewing Editor feels that the concerns raised during the last round of review were not addressed. Specifically, the reviewers and the editor argued that the presence or absence of currents cannot be used conclusively to determine the stoichiometry of transport. They have suggested that the corresponding section of the manuscript was rewritten in more speculative terms. However, the revised manuscript contains more definitive description of stoichiometry, which the Reviewing Editor deems inappropriate. Moreover, the new version seems to contain erroneous statements. For example, in one case the authors write: "[…]the apparent coupling stoichiometry changes specifically to 1:1 for aliphatic glucosinolates (no 4MTB over-accumulation, no induced currents)[…]". This does not seem to make sense. If there is 1:1 coupling, there should be concentrative transport and over-accumulation. We again urge the authors to rewrite the paragraph in less definitive more speculative terms and remove the corresponding stoichiometry table, which, in our opinion, over-interprets the data. If the authors disagree with us, we would like to hear their arguments as the letter accompanying the re-submitted manuscript does contain any discussion of the concerns raised by the reviewers and the Editor.

We ordinarily do not allow multiple rounds of review and revision thus this will be your final opportunity to provide an acceptable response to the Reviewing Editor's request.

---

## [Author Response]

*The reviewers found the work interesting, original and of considerable potential impact. High quality of experimental work was pointed out. Unfortunately, a number of significant issues have been identified that seem to be essential to for the conclusions that the authors draw.*

*Key issues:*

*1) The key point of the paper seems to be that Me14G074000 is the transition stage between transporters specific for cyanogenic gluocosides and glucosinlates. Could it be that Me14G074000 has an entirely different "native" substrate?*

Reviewer is correct in that Me14g74000 may transport other substrates and we acknowledge the need to explicitly state this in the text. We have added the following sentence: “It cannot be excluded that Me14g74000 also transports other substrates. An alternative substrate could explain why Me14g74000 is retained in the cassava genome.”

*It is odd that in a family of H-coupled transporters, there would be one that takes up relevant substrate, cyanogenic glucoside in a non-coupled manner.*

We agree that alternative explanations for the lack of currents are possible. Our first explanation suggested that cyanogenic glucoside binding alone is enough to induce conformational changes that flip the transporter and release the substrate into the cytoplasm. Alternatively, it could be speculated that a proton (or another cation) and an unidentified anion are co-transported with the cyanogenic glucoside. This alternative explanation is now included in the text:

“However, we cannot exclude that a pair of cations and anions are co-transported along with the cyanogenic glucoside, which would result in non-electrogenic yet coupled transport of cyanogenic glucosides. Thus, our data suggests that the change in substrate specificity from cyanogenic glucosides to glucosinolates was accompanied by a change in coupling stoichiometry between cyanogenic glucosides to cations, possibly from 1≤1 to 1:0.”

The authors suggest that cassava contains another transporter that would be specific for cyanogenic glucosides, while the Me14G074000 is a "transition" transporter. If so, a closely related specific transporter should be easily identified. The authors should attempt to find it.

We indeed proposed this hypothesis, where it reads:

“If Me14G074000 represents such a transition phase from cyanogenic glucoside to glucosinolate transport, we hypothesized that the genome of the cyanogenic glucoside-producing cassava also encodes a GTR-like NPF transporter that is specific for cyanogenic glucosides. We tested this hypothesis by measuring the transport activity of the six closest homologs of Me14G074000 from cassava (Figure 6B-C and Figure 6—figure supplement 2)”

We identify and characterize this transporter in Figure 6C-E. However, when we re-read this section we realize that this information may have been too convoluted within a large amount of data described. To make the findings and conclusion clearer we have reorganized the section to read as follows:

“If Me14G074000 represents such a ‘transition’ transporter between cyanogenic glucoside-specific to glucosinolate-specific transporter, we hypothesized that the genome of the cyanogenic glucoside-producing cassava also encodes a GTR-like NPF transporter that is specific for cyanogenic glucosides. […] Thus, the identification and close phylogenetic relationship of glucosinolate-specific, dual-specific and cyanogenic glucoside-specific transporters within the NPF strongly supports that glucosinolate transporters evolved from cyanogenic glucoside NPF transporters.”

*2) At least negative (preferably all) transport data need to be confirmed by expression analyses, such as a simple western and/or confocal analyses. This is in the transport field standard as absence of transport can have many reasons (such as no expression or miss-targeting etc.), while high transport rates can be simply caused by higher expression levels. For example, an important conclusion that the transporter from cacao does not transport glucosinolates is based on simple lack of activity, which is not convincing in the absence of expression analysis or a positive control, showing that this transporter transports its native substrate.*

To accommodate reviewer’s request we have YFP-tagged all transporters tested in figure 5 and 6 in the C-terminus and investigated expression using confocal microscopy. This data is included in two new figures: Figure 5—figure supplement 2 and Figure 6—figure supplement 1. The new data shows that expression was detected in all but two transporters among the cassava genes tested for cyanogenic glucoside transport in Figure 6. Our main conclusions, however, are based on transporters that are expressed. We have incorporated the new data in several locations in the text.

In subsection “Rise and evolution of glucosinolate transport specificity” we added the following text:

“To track the rise and evolution of glucosinolate substrate specificity we tested a range of transporters for glucosinolate transport activity via expression in X. laevis oocytes. All transporters described below were expressed and tested for transport activity in their native form. Additionally, each gene was fused to YFP in the C-terminus and its expression and localization to the plasmamembrane confirmed via confocal microscopy (Figure 5—figure supplement 2). In the following, lack of transport is therefore likely attributed to lack of activity rather than lack of expression.”

In the same subsection we added the following text:

“Oocytes expressing Me14G074000 from cassava over-accumulated both 4MTB and I3M relative to external media, while the expressed GTR homolog from cacao did not transport any glucosinolates (Figure 5B, Figure 5—figure supplement 3 and Figure 5—figure supplement 2).”

Subsection “Co-evolution of transporter substrate specificity and substrate biosynthesis” we added the following text:

”All six transporters were tested for transport activity in their native form. Additionally, we fused YFP to the C-terminus of each gene to validate expression.”

In the same subsection we added the following text:

“Among the YFP-tagged transporters only Me14G074100 and Me09G097200 did not express in the oocytes (Figure 6—figure supplement 1), and hence we cannot conclude whether these two transporters are inactive. Uptake of prunasin was detected in oocytes expressing Me15G176100 at levels similar to or slightly inferior to the suggested ‘transition’ transporter (Me14G074000) (Figure 6C).”

*3) The study includes phylogenetic analysis but omits any detail about specific residue changes between these transporters that could and should account for their biochemical differences. Given the recent structure of the NRT1.1 homologue and bacterial POT family members, the authors should examine the amino acid sequence changes between their characterized homologues and put these changes into structural perspective. They should discuss their findings in light of known conserved motifs.*

This issue is related to issue 6. We address them both in the answer under issue 6.

*4) The authors should interpret their results in structural perspective, perhaps mapping the amino acid differences on a structural model of the transporter and discuss potential substrate-binding site. We point out that in the NRT1.1 structural paper some of the key amino acids differing between NRT1.1 and distantly related bacterial peptide transporters were identified. Perhaps, these would be helpful in examining differences between GRTs and closely related transporters.*

To address reviewers’ comments we have constructed homology models using the recent structure of NPF6.3 as a template. In addition, we compiled a list of amino acid residues predicted to be key for substrate binding in structural studies on bacterial POT transporters and NPF6.3. The corresponding residues were located and numbered P1-P13 in an alignment comprising NPF6.3, POT transporters and the glucosinolate and cyanogenic glucoside transporters investigated in this study (Table 1 and Figure 8—figure supplement 1). We use the models and the alignment to discuss the measured substrate specificities in the current study in a structure function context. This is described in a new paragraph that we suggest to insert as a last paragraph before the conclusion:

“Structural perspectives on glucosinolate and cyanogenic glucoside substrate specificity

The large plant NPF family is homologous to the SLC15/PepT/PTR/POT families in bacteria and animals (Mansy et al., 2008,, Deamer and Dworkin, 2005). Several bacterial POTs (Weng, Philippe and Noel, 2012; Saier and Ren, 2006 and Ohno, 2013) and one plant NPF homolog (AtNPF6.3) (Fani and Fondi, 2009 and Prasad et al., 2012) have been crystalized along with their substrates. […] Our work provides a framework for future studies that seek to determine the amino acid changes that leads to substrate specificity changes during evolution of metabolite transporters.”

*5) The proton coupling mechanism of GTR3 in uptake of 4MTB is unclear. From the apparent lack of currents it can be concluded that the uptake is coupled to 1 proton, but the substrate concentrations used in this experiment (Figure 2A) are very low. How will the currents look like at higher 4MTB concentrations, where the uptake is more significant? Substrate-dependent currents should be examined and coupling mechanism should be established as it is very surprising that the mechanism seems to be 1:1 yet there is no accumulation in Figure 2D. Could the reversal potentials be used to better establish the coupling stoichiometry in these and other transporters examined in the study?*

We respectfully disagree with this comment stating that the substrate concentrations used in the experiment shown in Figure 2Ais very low. Actually it is high when compared to the Km. We determined the Km of AtGTR1 and AtGTR2 to be ~20µM towards both 4MTB and I3M (this study and (*10*)). Similarly, GTR3s Km towards I3M is ~20µM. Thus, the tested concentrations are 10-fold higher than the Km towards the known substrates. Furthermore, we see clearly defined substrate-induced currents (20% of maximal current) when the substrate concentrations are as low as 10µM for all three transporters. However, no currents are seen for GTR3 even at 200µM 4MTB. The glucosinolate concentrations measured in apoplastic fluid of wildtype Arabidopsis plants was previously estimated to be approximately 0.01 nmol/µl (*11*). Thus, at 200 µM we should be well above the physiological range of glucosinolate concentrations in the apoplastic space.

We previously explored the possibility of determining the reversal potentials as a means to determine the exact substrate to proton stoichiometry (supplemental data in (Khersonsky and Tawfik, 2010)). However, the theoretical reversal potential are calculated to be > +200mV, making it not feasible to use this approach.

*6) The finding that GTRL1/2 from Carica papaya transports both 4MTB and I3M is surprising and used as argument for the concept that glucosinolate transporter specificity arose from broad to narrow. Also, analysis of Me14G074000 from Cassava that transport both 4MTB and I3M supports this assumption. While we understand why these organisms have been chosen, we feel hat their individual choice and the number of tested transporters and species (here only 1 each) is too small to fully support these very important conclusions. We suggest including at least a second transporter from a similar but distant organism.*

We are happy that reviewer appreciates the experimental design and choice of organisms. We have in this study already shown that two transporters from Carica papaya (and not only one as indicated by reviewer) displayed broad specificity towards glucosinolates. In addition, we showed that a third transporter (Me14g74000) from a similar but distant (more basal) organism (cassava) also displayed broad specificity towards glucosinolates. We therefore believe that we already have provided the data requested by reviewer and that the substrate specificity of the selected transporters provide sufficiently strong evidence to support our claims. It should be noted that to generate these data we tested 6 homologs from *A. thaliana*, two homologs from B. rapa, 8 homologs from cassava, three from C. papaya and one from T. cacao (total of 20 genes tested).

Lastly, we would also like to emphasize that we are cautious in phrasing our claims regarding the evolution of the “narrowness” of substrate specificity towards glucosinolates. For example in of the manuscript:

“Thus, the data imply that the common ancestor of the GTR transporters was originally broad-specific and that GTR3 lost the ability to over-accumulate aliphatic glucosinolates after the divergence of C. papaya and the ancestor of Arabidopsis and Brassica (~ 72.1 MYA, median of 8 studies)”.

*7) A broader literature review is needed to discuss what is known in the field of evolution regarding development of novel specificities. This has previously been examined for hormone receptors, for example, in great detail.*

We have included a broadened literature review of what is known in the field of substrate specificity evolution with emphasis on the evolution of hormone receptors. We added the following paragraph:

“However, the evolutionary path that leads to the rise of new transporter substrate specificity upon emergence of new metabolites is unknown. Classical evolution theory (Halkier and Gershenzon, 2006) and several studies (e.g. (Bak, Nielsen and Halkier, 1998),(Clausen et al., 2015))) support the hypothesis that new enzyme functions arise in duplicated genes if they are subject to unique selection pressure, – alternatively they rapidly become pseudogenes. […] It is, therefore, not clear if new transporters evolve de novo with emergence of new substrates, or whether gene duplications allow ancestral multifunctional proteins to take on greater specificity (Züst et al., 2012).”

[Editors' note: further revisions were requested prior to acceptance, as described below.]

*After additional discussion, we feel that the authors have not sufficiently addressed the key issue 2) of the original review. We would argue that based on the data presented, it is not possible to make definitive statements about stoichiometry of transport. Specifically, we think that the presence of concentrative transport is clearly indicative of ion-coupled transport. Such concentrative transport is demonstrated for the uptake of glucosinolates by Me14G0176100 (Figure 5B) as well as by GTR1s and GTR3s (I3M). In contrast, transport of cyanogenic glucosides by Me14G0176100 is not concentrative (Figure 6 A and C). Thus, it is reasonable to hypothesize that it is not ion-coupled. However, the presence or absence of current is difficult to interpret based on the reported relatively preliminary electrophysiology experiments. For example, the observed current might be due to an uncoupled conductance that accompanies coupled transport and not due to stoichiometry of >2H^+^ per substrate. In that regard, we note that GTR1s and Me14G0176100 accumulate glucosinolates to a similar extent. Would not one expect GTR1s to concentrate substrates to a greater extent than Me14G0176100 if they operated with higher proton: substrate stoichiometry? Furthermore, the absence of current might be simply a consequence of slow rate of transport that would not allow detection of the ion flux. More detailed studies would need to be conducted to resolve these questions. We recognize that these go beyond the scope of the current manuscript. We urge the authors to re-evaluate their conclusions on the coupling mechanisms and to re-formulate them in more speculative, hypothetical terms.*

We thank the senior editor, reviewing editor and the reviewer for their positive comments and suggestions. We have reformulated the paragraph concerning the coupling mechanisms to highlight the more speculative and hypothetical nature of the data presented. We have included the new paragraph below in quotation marks.

“Are changes in transporter substrate specificity accompanied by changes in coupling stoichiometry?

Transporters of the SLC15/PepT/PTR/POT/NPF family couple substrate transport to the proton electrochemical gradient with varying coupling stoichiometry depending on substrate.[…] Future studies aimed at identifying the exact coupling stoichiometry for the cyanogenic glucoside and glucosinolate NPF transporters will unravel the mechanistic fine-tuning that accompanies evolution of substrate specificity.”

[Editors' note: further revisions were requested prior to acceptance, as described below.]

*The Reviewing Editor feels that the concerns raised during the last round of review were not addressed. Specifically, the reviewers and the editor argued that the presence or absence of currents cannot be used conclusively to determine the stoichiometry of transport. They have suggested that the corresponding section of the manuscript was rewritten in more speculative terms. However, the revised manuscript contains more definitive description of stoichiometry, which the Reviewing Editor deems inappropriate.*

We apologize for not explaining more specifically in our reply letter about our reason for writing as we did in the previous revised version of the manuscript. We also thank editor for providing us with an additional opportunity to address these concerns.

We agree with the reviewing editor’s opinion that the exact stoichiometry cannot be determined based on the presence or absence of currents. We used the term “apparent stoichiometry” and substituted show with suggests/indicate to convey the uncertain nature of the measurements. However, in hindsight we see that this did not accommodate reviewer and editor’s concerns. Consequently, we have now completely rewritten the paragraph in more speculative terms.

We now strictly refer to whether or not detectable currents where measured when a given substrate was transported by the different transporters in our study. Rather than inferring stoichiometries from these observations, the potential underlying reasons for why we see differences in detectable currents are now conveyed in what we believe are speculative terms that are subject to future studies (see below).

*Moreover, the new version seems to contain erroneous statements. For example, in one case the authors write: "…the apparent coupling stoichiometry changes specifically to 1:1 for aliphatic glucosinolates (no 4MTB over-accumulation, no induced currents)…". This does not seem to make sense. If there is 1:1 coupling, there should be concentrative transport and over-accumulation.*

This part and these speculations have now been removed from the paper. We agree that relating the absence of current for the transporter in question here (AtGTR3) to a given glucosinolate to proton stoichiometry is premature. We expect to address these interesting findings in future studies.

*We again urge the authors to rewrite the paragraph in less definitive more speculative terms and remove the corresponding stoichiometry table, which, in our opinion, over-interprets the data.*

We have removed:

– The stoichiometry table and references to it in the text

– The cation symbols and label from Figure 7 and edited figure legend

– Completely rewritten the paragraph previously talking about stoichiometries to avoid mentioning of deduced “apparent stoichiometries”. We report that the transport mechanism by which glucosinolates and cyanogenic glucosides are transported by the potential transition transporter Me14g074000 does not induce currents. This is in contrast to substrate transport by MeCGTR1 and At/Br/CpGTRs which all induce strong negative currents when exposed to their substrate. This indicates that there is a difference in the electrogenecity of the underlying transport mechanism. We then suggest that future studies should be aimed at unravelling whether the changes in electrogenicity are caused by changes in stoichiometry.

The new paragraph can be found in subsection “Is evolution of new substrate specificity in the NPF accompanied by changes in transporter electrogenicity?

We have further made changes that were required as a consequence of our changes to the rewrite of the paragraph discussed above as well as minor moderations of the Abstract to accommodate the 150 word max.

*If the authors disagree with us, we would like to hear their arguments as the letter accompanying the re-submitted manuscript does contain any discussion of the concerns raised by the reviewers and the Editor.*

We thank the reviewing editor for the chance to moderate our response and for the continued dialog discussing this very interesting observation. I hope that we now have satisfactorily addressed the concerns raised and look forward to hearing from you.